# Do LLMs Build World Representations?
# Probing Through the Lens of State Abstraction

**Zichao Li**[*]
Mila, McGill University
zichao.li@mail.mcgill.ca

**Yanshuai Cao**
Borealis AI
yanshuai.cao@borealisai.com

**Jackie C.K. Cheung**
Mila, McGill University
jackie.cheung@mcgill.ca

## Abstract

How do large language models (LLMs) encode the state of the world, including the status of entities and their relations, as described by a text? While existing work directly probes for a complete state of the world, our research explores whether and how LLMs abstract this world state in their internal representations. We propose a new framework for probing for world representations through the lens of state abstraction theory from reinforcement learning, which emphasizes different levels of abstraction, distinguishing between general abstractions that facilitate predicting future states and goal-oriented abstractions that guide the subsequent actions to accomplish tasks. To instantiate this framework, we design a text-based planning task, where an LLM acts as an agent in an environment and interacts with objects in containers to achieve a specified goal state. Our experiments reveal that fine-tuning as well as advanced pre-training strengthens LLM-built representations' tendency of maintaining goal-oriented abstractions during decoding, prioritizing task completion over recovery of the world's state and dynamics.[1]

## 1 Introduction

Drawing inspiration from human mental models [9, 12, 22], AI researchers have introduced the concept of world models for sample-efficient and robust machine learning systems [13, 25]. Specifically, a world model is defined to fulfill a dual role: (1) it estimates information about the world state that may not be directly observable from the input signals, and (2) it distills essential information to predict future states, thereby informing subsequent actions based on these predictions.

Recently, there has been growing interest in investigating whether pre-trained Transformer models [36], especially large language models (LLMs) [32], construct implicit world models. These investigations aim to determine whether the state of the world as described in the text, either implicitly or explicitly, can be recovered from the internal representations built by Transformer models. However, the field has produced studies with conflicting conclusions when examining different tasks. For example, [27] shows that it is possible to accurately probe the status of entities and their semantic relations encoded in the representations of LLMs within a discourse. Conversely, [24] report negative results when attempting a similar yet more challenging setting. In addition, [28] successfully extracts the board state of a partially played Othello game from the internal representation of a small-scale GPT model [7] trained to complete game scripts.

---

[*]Work was done during a Mitacs internship at Borealis AI.

[1]Code and dataset: https://github.com/BorealisAI/llm-world-abs

38th Conference on Neural Information Processing Systems (NeurIPS 2024).

We argue that these contradictions stem from the fact that recovering the complete world state is not always necessary for solving every tasks, which previous studies have not systematically controlled for. Rather, an abstract representation of the world state may sometimes be sufficient, and the necessary level of abstraction can also vary.

Some tasks may require the complete recovery of world dynamics in order to predict future states, while others may get by on one that omits this information as it is unnecessary for task completion. For instance, consider Figure 1. When parsing a discourse into a representation of the world state, one can either record the location of each object or simply note the number of objects in each box. The former offers the possibility to answer diverse questions, including future state predictions after op-

Figure 1: A discourse and two possible abstractions of the world state described by it (top half of figure). A general abstraction (termed as *world-irrelevant abstraction* later) enables one to answer a wide range of questions. On the other hand, a count-oriented abstraction is only applicable to the counting task.

erations, e.g., *Move the key to Box C*. The latter, however, is restricted to a counting task, unable to foresee future states. Neglecting this nuance could lead to a mismatch between evidence and conclusion, causing undue pessimism about LLMs lacking awareness of the world or excessive optimism regarding their ability to develop general world representations. For example, fine-tuning on all types of questions in Figure 1 might push LLMs to capture world dynamics, this does not guarantee the same outcome when fine-tuning solely on counting tasks.

We formalize this intuition through the application of *state abstraction theory*, originally proposed in reinforcement learning (RL) [29, 3] to simplify the state space by aggregating similar states to abstract ones without modifying the core aspects of the task or underlying world. The level of abstraction is a spectrum, from more general *world-irrelevant abstraction*, that allows recovery of world dynamics to more *goal-oriented abstractions*, guiding task completion while giving up on predicting future states. These include $Q^*$-irrelevant and $\pi^*$-irrelevant abstraction; the former preserves the long-term impact of actions, while the latter preserves the optimal policy.

Through this lens of state abstraction, we propose a new framework for examining the world representations constructed by LLMs. This framework investigates the various types of abstractions that may be encoded by LLM-built representations. To demonstrate its utility, we present a concrete application. First, we design a text-based planning task, REPLACE, which requires altering the state of a simplistic world with a collection of containers and objects. We intentionally craft the task's state space to be highly structured and modular, enabling a precise yet simple derivation of world state abstractions at different levels. Despite its simplicity, the abstract states at different levels are distinct and identifiable. Subsequently, we prompt pre-trained and fine-tuned LLMs to complete this planning task and extract their representation during decoding. Finally, we probe different abstract states within the representations. We conduct experiments on a wide range of Transformer models and LLMs, namely Pythia [6], Llama2 [35], Llama3 [1], Mistral [20] and Phi3 [2]. Our experiments show that LLMs achieving reasonable performance on REPLACE, whether through fine-tuning or advanced pre-training, tend to maintain goal-oriented abstractions rather than more general world representations during decoding. Additionally, pre-trained models with near-random performance fail to efficiently preserve any type of abstractions.

Our contributions are as follows: 1) We propose a new framework to probe world abstraction from LLM-built representations that can be adapted for other NLP tasks. 2) We release a new synthetic task, REPLACE, and accompanying datasets that are modular and extendable. 3) Experiments using our framework and task yield novel findings: LLM representations prioritize goal-oriented abstractions that preserve the effect of actions in terms of task completion while abstracting out the world state and dynamics during decoding. 4) Our findings also reconcile conflicting conclusions in prior work. For instance, [21, 28] successfully probe state variables like disc color and obstacle position, which

pertains to goal-oriented abstraction for the task that the models are optimized to solve, while [24] struggles to recover entity status, which lies outside these abstractions.

## 2 Related Work

### 2.1 World models

The concept of world models in machine learning has deep connections with the concept of human internal models [9] in cognitive science, most often referred to as mental models [12, 22]. As originally defined, mental models can build abstract and symbolic representations of entities and their relations in the real world or environment around them [12]. One of the most important characteristics of such mental models is their optimal balance between representation complexity and utility for accomplishing a specific task [18]. In a similar vein, AI researchers have developed symbolic [33] or neural world models [13, 14] to compress vast amounts of input information and extract a simplified and essential representation to predict future states. Our work is related to both domains and applies state abstraction theory [29, 3] from reinforcement learning to assess the representations of the underlying world and the task, if any, encoded by LLMs.

### 2.2 Probing LLM Representations

There is a growing interest in probing for interpretable features in LLMs, most of which are linguistic in nature, such as morphology [30], syntax [17] and word-level semantics [5]. Recently, researchers have moved beyond the exploration of shallow linguistic features, investigating whether and how LLMs' hidden representations on the fly encode entities and their relations, as described in the text. More specifically, [27] trains a shallow neural classifier on top of LLMs' representations of discourse to recover the ground-truth situations as depicted in the text. This approach is further modified and formulated as a next-sentence prediction task by [24]. Similarly, [28, 21, 37] probes the internal representation of world state in Transformer models reading (semi-)structured input, such as game scripts and embodied sequences.

However, existing work primarily focuses on probing for a comprehensive description of the underlying world, defined as a set of state variables. This approach has two main limitations. First, it fails to distinguish the function of a specific state variable: Is it intended to maintain a general representation of the world, or is it crucial only for specific tasks, or both? Without this distinction, we cannot accurately interpret the positive outcomes of probing. Second, existing studies do not account for the potential abstraction of the world that LLM might build in its representation. Our work overcomes these limitations by introducing a general framework that examines the world abstraction encoded within LLM representations.

In contrast to [28, 21, 37], the input in our task is text data instead of a game script or environment layout. Therefore, the conclusions drawn from our experiments are more applicable to LLMs and Transformer models for NLP tasks.

## 3 Framework

In this section, we introduce a new probing framework to investigate the world state abstraction encoded by large language models (LLMs) when prompted to perform a decision-making task. We start with a reinforcement learning (RL) formulation, which is general enough to encompass a wide range of NLP tasks and other tasks adopted by previous work, e.g. completing an Othello game script [28]. An RL problem is characterized by a 4-tuple: $(\mathcal{S}, \mathcal{A}, T, R)$, where $s \in \mathcal{S}$ represents the world state, $a \in \mathcal{A}$ denotes possible actions the model can take, transition function $T(s'|s, a)$ measures the probability of transitioning to state $s'$ induced by $a$ from $s$, and $R(s, a)$ is the reward.

### 3.1 Definition and Derivation of World State Abstraction

Our framework builds upon a rigorous definition of *world abstraction*. In particular, we follow RL literature [29], deriving state abstraction function $\phi : \mathcal{S} \to \mathcal{S}_\phi$ that maps each state $s$ into an abstract state $\phi(s)$ ($|\mathcal{S}_\phi| \leq |\mathcal{S}|$). In addition to the raw state $\mathcal{S}$, we consider three types of abstraction:

**World-irrelevant abstraction** $\phi_w$ [2] ensures that $\forall s_1, s_2$ where $\phi_w(s_1) = \phi_w(s_2)$ implies that $\forall a, x' \in \mathcal{S}_{\phi_w}, R(s_1, a) = R(s_2, a)$, and $\sum_{s' \in \phi_w^{-1}(x')} T(s'|s_1, a) = \sum_{s' \in \phi_w^{-1}(x')} T(s'|s_2, a)$ . Intuitively speaking, it preserves the transition dynamic of the underlying world and the reward function, thereby enabling the prediction of future states induced by subsequent actions. Recalling the definition of world models in Section 1, this type of abstraction enables precise recovery of the world model.

$Q^*$-**irrelevant abstraction** $\phi_Q$ ensures that $\forall s_1, s_2$, if $\phi_Q(s_1) = \phi_Q(s_2)$, then $\forall a, Q^*(s_1, a) = Q^*(s_2, a)$, where $Q^*(s, a) = \max_\pi Q_\pi(s, a)$. This form of abstraction preserves the effect of all actions $a \in \mathcal{A}$ in terms of their optimal $Q$-value, which is the maximal expected future rewards. However, it discards the world dynamics, making future state predictions infeasible. LLMs can leverage this abstraction to be cost-sensitive, minimizing action counts and avoiding penalties for violating world constraints.

$\pi^*$-**irrelevant abstraction** $\phi_\pi$ guarantees that the optimal action, and hence the optimal policy, can be recovered. Formally, $\forall s_1, s_2 \in S$ , $\phi_\pi(s_1) = \phi_\pi(s_2)$ implies that $\pi^*(s_1) = \pi^*(s_2)$. Therefore, this is the coarsest abstraction that one can recover the optimal policy. Therefore, we expect that at least this type of abstraction can be accurately probed from an LLM that excels in the given task.

In practice, one may use learning [23, 4] or heuristics [10, 11] to derive the abstractions at each level.

### 3.2 Probing World Abstraction from LLM-built Representations

We aim to assess which types of state abstraction are encoded in LLM representations. To do so, we first prompt a pre-trained or fine-tuned LLM with $\tau(s_0, A_{1:t-1}, z)$, which is a textual description of the initial state $s_0$, previously executed actions $A_{1:t-1}$, and optionally, feedback $z$ received from the world. The feedback could be the opponent's moves in an Othello game or users' utterances in a dialogue system. Next, we extract the hidden states $H^{(m)}$ from the $m$-th layer of the LLM, considering $m$ as a hyperparameter. Following [28], we select the last hidden states $h_t^{(m)}$[3], which is used by the LLM to predict subsequent actions. After collecting $(s_t, h_t)$ pairs, where $s_t$ is the world state induced by $s_0$, $A_{1:t-1}$ and $z$, we detect the existence of abstraction of $s_t$ in LLM representations by assessing if $\phi(s_t)$ can be probed from $h_t$ with accuracy surpassing a random baseline, potentially achieving near-perfect performance. Previous work [16, 17] on probing presents a challenge in differentiating whether a representation encodes a linguistic property or if the probe itself learns the task. While these works focus on low-level syntax features [16, 30], our focus is on the recoverability of world states and their abstractions from LLM representations, questioning whether these are maintained or discarded during decoding. Therefore, we train probes to classify raw and abstract states, following previous work [27, 28], which essentially estimates the mutual information between abstractions and representations [34]. As we will show in Section 6, probing either raw or abstract states from LLM representations is not always successful.

**Remark:** To draw faithful conclusions from probing experiments, it is crucial to carefully design or select the tasks in a way that ensures the spaces of abstract states at different levels differ ($\mathcal{S}_{\phi_w} \neq \mathcal{S}_{\phi_Q} \neq \mathcal{S}_{\phi_\pi}$). Otherwise, one cannot determine whether the success of probing stems from the LLM's preference for learning a general world model or from the necessity to recover the world state while learning the optimal policy. For instance, in the game of Othello, it is feasible to recover most of the current board state from all possible legal moves. As such, the raw state space is almost identical to the coarsest $\pi^*$-irrelevant abstraction for predicting legal moves. Therefore, a plausible interpretation for the success of probing in [28] could be that the Transformer model learns the $\pi^*$-irrelevant abstraction rather than deliberately learning a general world model.

In the next section, we design a planning task within an RL framework and synthesize a dataset accordingly. In Section 5, we prompt LLMs to perform this task and probe different types of abstractions from their internal representations.

## 4 REPLACE: A Text-based Planning Task

To instantiate our framework, we draw inspiration from recent work on probing discourse representation in LLMs [27, 24] and design a text-based planning task named REPLACE. In REPLACE,

---

[2]It is originally referred to as *model-irrelevant abstraction* [29]. Here, we rename it to avoid confusion between world models and large language models.

[3]We will omit the layer index in the following sections for simplicity.

*{{Task Instruction}}*
*A sequence of containers are ordered from left to right as follows: Bucket A, Basket A, Basket B, Crate A, Basket C. You are at the Basket B.*
***Initial situation***: *Bucket A is occupied by a car, a key. Basket A contains a creature, a letter. Basket B has a beer. Crate A holds a coat. Basket C holds a crown.*
***Desired situation***: *Bucket A contains a car. Basket A is occupied by a creature. Basket B is empty. Crate A holds a beer, a coat. Basket C has a crown.*
***Operations applied***: *You Move left.*
*What are the operations to achieve the desired state?*

Figure 2: An example of task input in REPLACE.

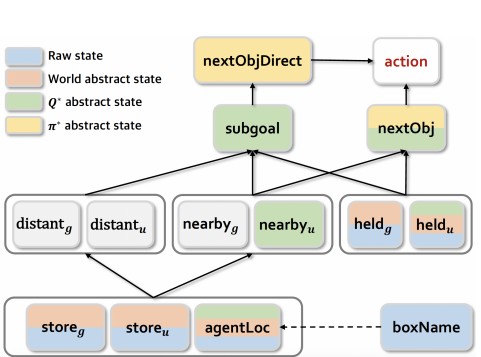

Figure 3: Derivation of abstract predicates and action from raw predicates. Better viewed in color. Predicates represented by multicolored squares indicate association with multiple abstraction levels. Grey squares represent intermediate predicates, not associated with any abstraction, used to derive abstract predicates.

| boxName | [*Bucket A*, *Basket A*, *Basket B*, *Crate A*, *Basket C*] |
|---|---|
| agentLoc | 2 |
| store$_u$ | [{*car*, *key*}, {*creature*, *letter*}, {*beer*}, {*coat*}, {*crown*}] |
| store$_g$ | [{*car*}, {*creature*}, {}, {*coat*, *beer*}, {*crown*} ] |
| distant$_u$ | [{*car*, *key*}$_{-1}$, {*beer*}$_{+1}$, {*coat*}$_{+2}$, {*crown*}$_{+3}$] |
| distant$_g$ | [{*car*}$_{-1}$, {}$_{+1}$, {*coat*, *beer*}$_{+2}$, {*crown*}$_{+3}$ ] |
| nearby$_u$ | {*creature*, *letter*} |
| nearby$_g$ | {*creature*} |
| held$_u$ | {} |
| held$_g$ | {*key*, *letter*} |
| subgoal | [0, −1, +1, +2] |
| nextObj | *letter* |
| nextObjDirect | nearby-container |

Figure 4: The raw and abstract predicates of the state described in Figure 2. $u, g$: current and target situations.

an LLM acts as an agent within a simple world consisting of a set of containers and objects. The objective of the LLM is to predict actions that alter the situation of the objects to match a described target situation. We design this task for three reasons. First, in its symbolic form, the task possesses a simple and modular state structure, making it easier to analyze and derive abstract state spaces. Second, it is closely related to the gripper problem [31] and the entity tracking task [24], which are widely adopted in planning and NLP interpretability research. Third, as we will demonstrate, each type of abstraction has a unique state space, enabling the assessment of whether LLM-built representations encode specific abstractions.

In this section, we start with formulating the planning task within the RL framework, then introduce the textual realization of the RL elements, and finally the method of curating prompts.

## 4.1 Specification within RL Formulation

Formally, the specific design of the elements within the RL framework is described as follows:

**1.** State $s = [u, g] \in \mathcal{S}$: the current situation $u$ and target situation $g$ of the world, each of which is factorized as the Cartesian product of a set of assignments over predicates applied to entities **e**, including objects **o** and containers **b**. The predicates are:

- store$(i, o)$: the object $o$ is stored in the $i$-th container from left to right.
- held$(o)$: the agent has the object $o$.
- agentLoc$(i)$: the agent is at the $i$-th container.
- boxName$(i, b)$: the name of the $i$-th container from left to right is $b$.

**2.** The possible actions $a \in \mathcal{A}$ include:

- move$(d)$: Move along a direction $d$, left or right, for one step.
- grab$(o)$: Grab an object $o$ from the nearby container.

- `put(o)`: Put an object from the agent to the nearby container.

**3.** Transition function $T$: Following previous work [27], we consider a static environment. That means $T(s'|s, a) \in \{0, 1\}$. Moreover, the transition impacts only the current situation $u_t$ in $s_t$ by taking the effect of $a$. For example, if `agentLoc(2)` is `True` in $u_{t-1}$, executing the action `move(left)` results in `agentLoc(1)` becoming `True` in $u_t$.

**4.** Reward function $R(s, a)$ checks whether the new situation of **o**, induced by executing $a$, matches the target situation: $R(s, a) = r$ if $\mathbb{P}_\mathbf{o}(u') = \mathbb{P}_\mathbf{o}(g)$; or $-(r + r')$ if $a \notin C(s)$; or $-r$ otherwise. Here $u'$ denotes the new situation, $\mathbb{P}_\mathbf{o}(u')$ the set of assignments on predicates involving objects **o** (`store` and `held`) in the new situation $u'$, and $C : \mathcal{S} \to \mathcal{A}$, the constraint function mapping states to permissible actions. We incorporate three spatial constraints: (1) the agent cannot move `left` and `right` at the leftmost and rightmost locations, respectively; (2) `grab` is only possible if the target object $o$ is in a nearby container; (3) `put` is allowed only for objects currently held by the agent. Given the LLM's insensitivity to exact numerical values in prompts, we do not set concrete values for $r$ and $r'$ but simply assume $r > 0$ and $r' > 0$. Notably, both $Q^*$ and $\pi^*$, as well as the abstract states, are invariant to the actual magnitudes of $r$ and $r'$.

We map the situation ($u$ or $g$), previous actions ($A_{1:t} = \{a_1, .., a_t\}$) to their respective natural language descriptions $\tau_s(u)$, $\tau_s(g)$, $\tau_a(A_{1:t})$, and using a set of textual templates: $\tau_s(\cdot)$ and $\tau_a(\cdot)$. As such, the input text to the model comprises two main components: (1) a general task instruction (see Appendix C) specifying the actions the agent can take, $\{\tau_a(a)|a \in \mathcal{A}\}$, and the constraints it must adhere to, $\tau(C)$, and (2) textual descriptions of the world's initial and target situations, as well as previous actions $[\tau_s(u); \tau_s(g); \tau_a(A_{1:t})]$. An example is presented in Figure 2. Conditioned on this, the LLM generates textual descriptions $\hat{Y}$ of subsequent actions, which are parsed back into their symbolic form $\hat{A}_{t+1:T}$ by a rule-based parser $\tau_a^{-1}(\hat{Y})$.

## 4.2 Datasets

We synthesize two English datasets for REPLACE: GRIPPER and COOK. For each instance in GRIPPER, we first sample a set of **o** from a list of container names and **b** from a list of frequent nouns in British National Corpus (BNC) [26]. Subsequently, we generate pairs of initial and target situations, $[u_0, g]$, by randomly assigning values to the predicates in $u_0$ and $g$, such that it requires the execution of 2 to 6 actions to transition from $u_0$ to $g$. Towards a more realistic setting, we introduce the following dataset variants: (1) *Lexical variants*, which includes rare nouns collected from BNC for **b**, adding color and size modifiers to describe the objects, and use diverse textual templates for translating predicates; (2) *Partial $\tau(g)$*, which omits explicit information about `held` in the target situation, requiring logical inference by LLMs; (3) *Partial $[\tau(u_t), \tau(g)]$*, an extension of (2) that includes previous operations performed by the agent, thus requiring inferring $u_t$. Appendix C provides concrete examples for each variant. The final dataset is a uniform mixture of these four variants and segmented into training, validation, and test sets, with $[u_0, g]$ splits, allocating 50k, 1k, and 1k instances to each set, respectively.

COOK generalizes the setting of GRIPPER to a grid world of containers and is generated with TextWorld [8]. Similarly, we sample and transform the initial and goal situations into textual descriptions. The main differences between COOK and GRIPPER are: (1) the constants of direction $d$ are different: the agent in COOK can move `north`, `south`, `east` or `west`; (2) COOK uses a different set of textual templates adopted from the *cook* domain in TextWorld. See Appendix D for more details about COOK. Notably, both datasets share the same world abstractions at all levels.

# 5 Probing World Abstractions in LLM Representations using REPLACE

For each LLM prompted to solve REPLACE, we assess which types of abstractions of state $s_t$ are preserved in its representation $h_t$ during decoding after the collection of $\{(s_t, h_t)\}$ pairs as described in Section 3.

The modular state structure in REPLACE allows us to analytically derive a set of predicates for each type of world state abstraction. To begin with, we introduce a set of abstract predicates crucial for devising the optimal plan for task completion:

- `nearby(o)`: the object $o$ is located within the container nearby to the agent. Similarly, we have `distant(l, o)`, where $l$ is the relative distance to the agent's current location, e.g., `distant(−1, o)` being `True` indicates that the container one step to the left contains $o$.

- subgoal$(l, j)$: the agent needs to visit a series of containers to manipulate objects, and the $j$-th container to be visited is at the relative distance $l$ to the agent's current location.
- nextObjDirect$(v)$: the relative direction of the next object to be manipulated, where $v$ could be left or right to the agent, in a nearby container, or with the agent.
- nextObj$(o)$: the object $o$ to be manipulated at the next step, if applicable.

Appendix E provides a detailed explanation of the definition and derivation of these predicates. Finally, we can infer the optimal action for the next step by incorporating nextObj and nextObjDirect. Figure 3 demonstrates how to derive abstract predicates and actions based on the raw predicates. Figure 4 provides concrete examples of predicates.

Now, we derive each type of world abstraction, using both raw and abstract predicates:

**Raw state** The complete world state $\mathcal{S}$ includes store, held, agentLoc in both the current and target situations, $u$ and $g$, and boxName.

**World-irrelevant abstraction** To preserve the world dynamics, we have to track all raw predicates of the current situation, excluding boxName, as actions do not affect it. In addition, we include all predicates in target situation $g$ to preserve the reward function. Therefore, $\mathcal{S}_{\phi_w}$ includes the predicates store, held and agentLoc in both $u$ and $g$.

$Q^*$**-irrelevant abstraction** To construct $\mathcal{S}_{\phi_Q}$, we include agentLoc, held$_u$ and nearby$_u$ to distinguish between the effects of legal and illegal execution of actions. Also, we include subgoal, where subgoal$(l, j = 1)$ differentiates the effect between optimal and non-optimal actions, and subgoal$(l, j > 1)$ quantifies the effect of executing each action. Further, we include nextObj to distinguish the effect between optimal and non-optimal put or grab actions.

$\pi^*$**-irrelevant abstraction** $\mathcal{S}_{\phi_\pi}$ is factorized by predicates nextObjDirect and nextObj. nextObjDirect specifies the action type (move or manipulate objects) and moving direction (left or right) at the next step. nextObj identifies the particular $o$ for grab or put operations.

Appendix G provides a more formal proof.

We probe the predicates in LLM representations by training two-layer neural models, following [28]. For single-variable predicates (e.g., nearby$(o)$), we train an individual probe for each. For two-variable predicates where one variable denotes a positional index (e.g., boxName$(i, b)$), we train a separate probe for each $i$-th position. For some predicates, e.g., store$(i, o)$, involving variables $o$ with domains that differ across the dataset, the input to probes is a concatenation of $[h_t, \varphi(o)]$, where $\varphi(o)$ is the embedding of $o$. Following the conditional probing principle [15], we treat the embedding method for each predicate as a hyper-parameter, selecting the method with the largest performance margin between using $[h_t, \varphi(o)]$ and using only $\varphi(o)$ as the input. Specifically, we explore two embedding methods: (1) averaging the word embedding for all the tokens in $o$ and (2) selecting and averaging the hidden states $H$ across all mentions of $o$. For other predicates, such as agentLoc and subgoal, probes take as input $h_t$. Appendix H provides more details about the formulation of probing tasks as well as the design (e.g. layer number) and optimization of probes.

**Evaluation metric** We evaluate the performance of probing models for each predicate using the F1-score. To account for variations in candidate numbers across different predicates, we normalize the scores $x_p$ for predicate $p$ by $\frac{\max(x_p - \beta_p, 0)}{1 - \beta_p} \times 100\%$, where $\beta_p$ is the score of a RANDOM baseline. It predicts outcomes by proportional sampling from label candidates of $p$ based on their frequencies. We refer to this normalized score as the *recovery rate*.

## 6 Experiments

### 6.1 Target Models for Probing Experiments

We experiment with two groups of LLMs to investigate how fine-tuning and pre-training impact the world abstractions encoded in their representations, respectively. The details follow below.

**Effect of fine-tuning** We adapt Llama2-7b/13b [35], Mistral-7b [20], and LLama3-

| Models | GRIPPER | | | COOK | | |
|---|---|---|---|---|---|---|
| | %Legal | %Succ | %Optim | %Legal | %Succ | %Optim |
| Llama2$_{\mathrm{ICL}}^{13b}$ | 62.41 | 3.07 | 0.19 | 35.41 | 1.70 | 0.36 |
| Mistral$_{\mathrm{ICL}}$ | 41.12 | 2.12 | 0.65 | 44.41 | 1.95 | 0.22 |
| Llama2$_{\mathrm{SFT}}^{13b}$ | 96.54 | 88.30 | 84.02 | 95.71 | 85.11 | 69.17 |
| Mistral$_{\mathrm{SFT}}$ | 97.07 | 92.15 | 87.36 | 95.04 | 85.91 | 61.47 |

Table 1: Planning performance of Llama2-13b and Mistral on GRIPPER and COOK.

8b [1] on both datasets, using in-context learning (ICL) and supervised fine-tuning (SFT) with LoRA adapters [19]. Appendix F provides more details.

**Effect of pre-training** Since the LLMs above perform near-random without fine-tuning, we also experiment with a more recent state-of-the-art LLM, Phi3-17b [2], which achieves much higher performance than other pre-trained LLMs on GRIPPER[4]. For comparative analysis, we also include Phi3-3.8b, Pythia-70m [6] and a fine-tuned Phi3-17b.

In addition, we train a 6-layer decoder-only Transformer from scratch, probing for abstractions from its internal representations as a baseline.

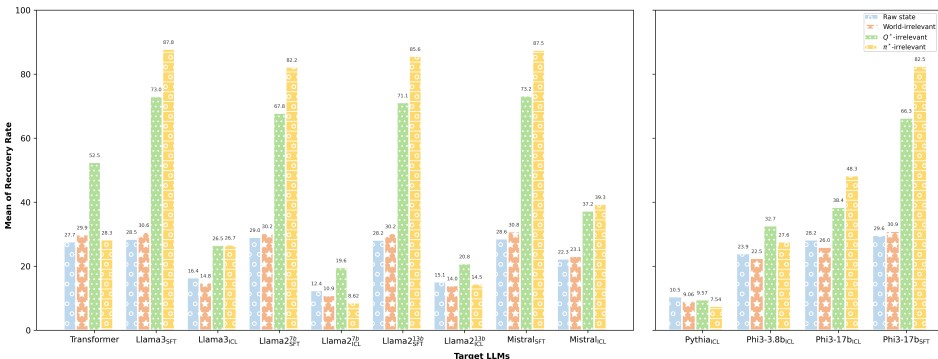

Figure 5: Average recovery rate of each world abstraction across different LLMs on GRIPPER.

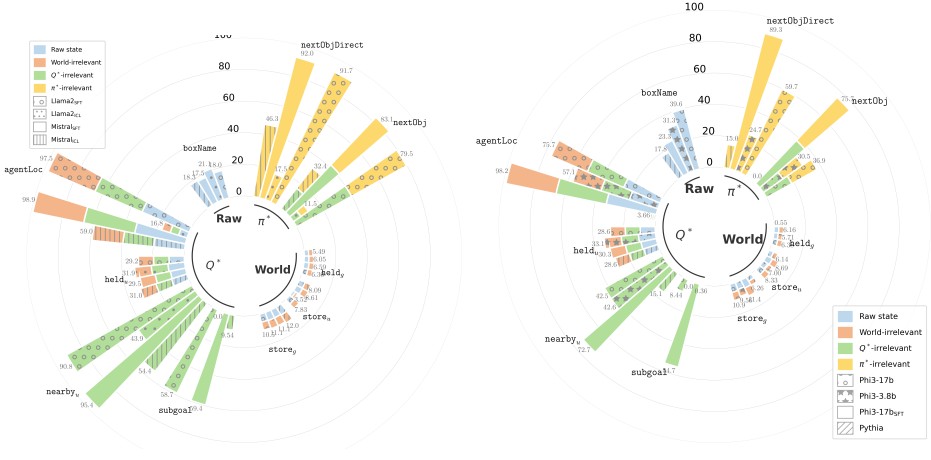

(a) From Llama2 and Mistral.       (b) From Pythia and Phi3.

Figure 6: Recovery rate of all predicates from different LLMs on GRIPPER. Predicates are grouped according to the coarsest abstractions they pertain to. Better viewed in color. The color indicates all abstraction levels the predicates are associated with.

Before the probing experiments, we first verify that the fine-tuned LLMs and Phi3-17b_ICL can achieve reasonable performance on REPLACE, and thus, we can draw informative conclusions. We adopt three evaluation metrics for REPLACE: legal rate (the proportion of predictions that comply with constraints), success rate (the proportion of predictions that achieve the target situations), and optimal rate (the proportion of cases where the target situation are achieved with the minimum number of actions). The Llama2-13b and Mistral results are reported in Table 1, with comprehensive results for all LLMs provided in Appendix I. As shown in Table 1, both LLMs initially fail in nearly all cases when relying solely on in-context demonstrations. Once fine-tuned, however, they can accomplish the task with reasonable success and optimal rate, and most of the predicted actions adhere to the constraints. Next, we conduct probing experiments on each variant of the LLMs on the same datasets.

---

[4]We do not conduct probing experiment with the pre-trained Phi3-17 on COOK as it does not outperform the random baseline significantly.

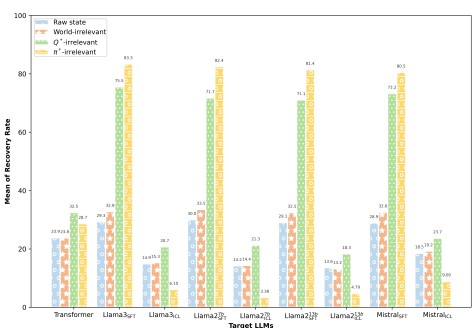
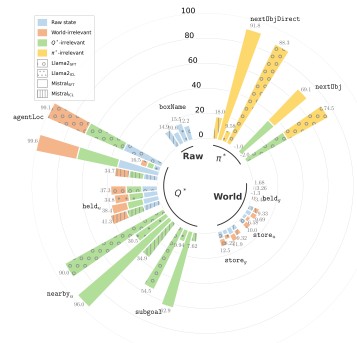

(a) Average recovery rate of each world abstraction.

(b) Recovery rate of all predicates from Llama2-13b and Mistral.

Figure 7: Recovery rate of predicates within different world abstractions across different LLMs on COOK.

## 6.2 Probing Results and Analysis

We measure probes for all the LLM variants adapted to REPLACE (ICL or SFT) as well as the train-from-scratch Transformer. Figure 5 and Figure 7(a) report the average recovery rate (RR) of predicates within each type of world abstractions from different LLMs. Figure 6 and Figure 7(b) take a closer look at the RR of each predicate from seven LLM variants to analyze the effect of fine-tuning and pre-training. The complete results are reported in Appendix J. Next, we present four key findings from the results.

**Finding 1.** *Reasonably performing LLMs tend to maintain goal-oriented world abstractions rather than a more general one during decoding.*

All LLMs$_{\text{SFT}}$ and Phi3-17b$_{\text{ICL}}$, which outperform the random baseline by a substantial margin, primarily preserve $Q^*$- and $\pi^*$-irrelevant abstractions. Comparing LLMs$_{\text{SFT}}$ and LLMs$_{\text{ICL}}$ in Figure 5 and Figure 7(a), $Q^*$- and $\pi^*$-irrelevant abstractions are probed from LLMs$_{\text{SFT}}$ with drastically higher RR than raw and world-irrelevant abstraction. Focusing on the RR of individual predicates, Figure 6(a) and Figure 7(b) reveal that the predicates for $\pi^*$-irrelevant abstraction, nextObj and nextObjDirect have been mostly recovered from LLMs$_{\text{SFT}}$. This result aligns with the high task success rate achieved by these LLMs. Similarly, the predicates for $Q^*$-irrelevant abstraction are effectively probed from the LLMs$_{\text{SFT}}$, although with larger variance across different predicates. This suggests that LLM representations do not merely encode the most coarsest world abstraction for deriving the very next action. Instead, they preserve sufficient information to estimate each action's long-term effect with respect to altering the world situation to match the target, facilitating efficient planning. The successful recovery of nearby, held$_u$ and agentLoc indicates that LLM representations capture the world constraints to inform legal actions, as evidenced by the near-perfect legal rate of LLMs$_{\text{SFT}}$.

Nevertheless, Figure 5 shows that although the RR margin between goal-oriented abstractions and more general ones is smaller for Phi3-17b$_{\text{ICL}}$, it remains significant. Figure 6(b) explains this by revealing that the RR for nextObj, nextObjDirect, and subgoal from Phi3-17b$_{\text{ICL}}$ is much lower than that of LLMs$_{\text{SFT}}$, which is unsurprising as Phi3-17b$_{\text{ICL}}$ achieve a much lower success rate on REPLACE than LLMs$_{\text{SFT}}$.

Conversely, more general world abstractions are mostly absent in all LLM representations. Predicates uniquely tied to the raw state and the world-irrelevant abstraction cannot be accurately recovered. The low RR for boxName suggests that LLMs discard world details that are not pertinent to the task's completion. Similarly, the minimal RR on store$_t$, a key predicate for reconstructing the world model, implies that the information to derive world dynamics is mostly omitted during decoding.

We create two other variants of GRIPPER to further consolidate this finding. The first one is a counterfactual setting where a predicate originally irrelevant to the goal becomes pertinent to goal-oriented abstraction. Our results show that it is accordingly encoded in the LLM$_{\text{SFT}}$ representations. Under the other setting, the LLMs are fine-tuned on sub-optimal action sequences, and the results suggest that random exploration does not necessarily improve the encoding of world dynamics. Details of these experiment can be found in Appendix K and Appendix L, respectively.

**Finding 2.** *Supervised fine-tuning and advanced pre-training mainly enhance goal-oriented world abstractions.*

Figure 6 and Figure 7(b) demonstrate significant differences in the recoverability of goal-oriented abstractions from representations of LLMs, comparing models with and without supervised fine-tuning. Specifically, the performance gap in terms of predicates `nextObjDirect`, `agentLoc`, and `subgoal` is notable, approximately $40 - 70$ recovery rate. In contrast, for `boxName` and `store`, the difference is much more negligible, less than $5$. This disparity implies that when fine-tuned with teacher forcing, LLMs evolve to develop a more goal-oriented world abstraction during decoding. This finding also provides a novel perspective on [21, 28] that successfully probe the world state in Transformer models: the state information is an integral part of the goal-oriented abstraction in their task, which is enhanced as the models are optimized to solve the task.

Interestingly, advanced pre-training has a similar effect. Figure 5 and Figure 7(a) show that the key distinction between train-from-scratch Transformer and LLMs$_{\text{SFT}}$ lies in their retention of $Q^*$- and $\pi^*$-irrelevant abstractions. Nonetheless, as LLMs increase in scale and capability (Pythia<Phi3-3.8b<Phi3-17b), they are more likely to maintain goal-oriented abstractions over a more general one. This is apparent from Figure 6(b), where the RR of predicates does not increase with Phi3-17b unless the predicates pertains to the goal-oriented abstractions. In contrast, the RR of `store` and `held`$_g$, essential for world-irrelevant abstraction, are almost identical across all pre-trained LLMs. It implies that advanced pre-training does not necessarily lead to better encoding of the world dynamics.

**Finding 3.** *Raw state and world-irrelevant abstractions are mostly suppressed during decoding.*

Motivated by our initial two findings, we explore whether and how `boxName` and `store` are encoded within the contextualized representation by Llama2$_{\text{SFT}}^{13b}$. To do so, we use the contextualized embedding Ctxt($\cdot$) of the label candidate, comparing the use of only the embedding Ctxt($e$) against a concatenation $[h_t; \text{Ctxt}(e)]$, in-

| Predicates | Abstraction Type | $[h_t; \text{Ctxt}(e)]$ | Ctxt($e$) |
|---|---|---|---|
| boxName | Raw | 100.00 | 100.00 |
| store$_u$ | Raw, World-irrelevant | 93.32 | 95.71 |
| store$_g$ | Raw, World-irrelevant | 93.01 | 95.18 |
| nearby$_u$ | $Q^*$-irrelevant | 92.11 | 42.46 |

Table 2: Recovery rate with different encoding methods of label candidates.

spired by [15]. We also examine `nearby` for comparative analysis. The results in Table 2 show that the objects' location can be probed from the contextualized representation with near-perfect performance while incorporating $h_t$ may decrease the performance instead. As for more goal-oriented predicates, `nearby` cannot be probed from the contextualized representation with high accuracy, and the concatenation with $h_t$ boosts the performance vastly, suggesting that LLMs discard the information of `store` and integrate it with `agentLoc` to derive `nearby`.

**Finding 4.** *LLMs are limited in building world representations, whether general or goal-oriented.*

While LLMs tend to preserve a goal-oriented abstraction, they do not totally discard `boxName` (15-20 RR from LLMs$_{\text{SFT}}$ and even 43 from Phi3-17b), which is irrelevant to either task completion or transition dynamics. Also, the probing performance for some goal-oriented predicates, e.g., `subgoal`, has ample room to improve. Moreover, the information for world dynamics, e.g., `store`, is mostly discarded.

**Remark.** *The interpretation of probing performance for a predicate should depend on the specific type of abstraction it belongs to.*

Focusing solely on raw world states might lead to a superficial conclusion that the LLM only partially captures the world. In contrast, our framework offers a more thorough and systematic interpretation, demonstrating that well-performing LLMs maintain a goal-oriented world representation during decoding. ***This underlines the core rationale of our probing framework: the necessity of probing different world abstractions rather than just the raw state.***

## 7 Conclusion

We propose a new framework to probe the abstract world state in LLM representations through the lens of state abstraction. Experiments with a synthesized task using our framework demonstrate that LLMs tend to preserve a goal-oriented world abstraction instead of a more general one during decoding, abstracting away the world's transition dynamics. Overall, our experiment findings highlight the importance of probing different types of abstraction encoded in LLM representations to draw comprehensive and nuanced conclusions. Nevertheless, our work is not without limitations, which are discussed in Appendix A.

## Acknowledgements

This work is supported by the Mitacs Accelerate Program (Project ID: IT27067) and partially enabled by Mila's computing resources (mila.quebec). We thank Raquel Aoki from Borealis AI for discussions, and Michael Rizvi, Maxime Darrin, and Saba Ahmadi from Mila for their manuscript feedback during paper swap. Finally, we appreciate the anonymous reviewers for their comments on clarity and suggestions for additional experiments.

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

## A    Limitations

**Limitations of our framework**    Our framework primarily investigates the presence of different abstraction types in LLMs, without delving into the mechanisms through which LLMs generate or link these specific abstractions. We leave these topics for future research. Additionally, as with many probing methods [34, 16], we cannot ensure that LLMs accurately and effectively make use of the identified abstraction. Our research instead focuses on whether it is possible to recover certain types of abstractions from LLM representations.

**Limitations of our designed task**    While the task we designed in this work allows for precise and analytical derivation of world abstractions and involves fundamental skills essential for other NLP and reasoning tasks, it is important to note that it remains relatively simple and artificial compared to real-world NLP tasks. Consequently, future work can extend this task to more realistic and complex settings.

**Limitations of our datasets**    Following previous works in this line of research [24, 28], we synthesize datasets for our task. Although synthetic datasets offer precise control over their properties and the characteristics of the underlying world, there may be discrepancies between our datasets and those used in real-world NLP applications. Unfortunately, realistic NLP datasets paired with a "world simulator"—vital for tracking state updates triggered by predicted actions in probing experiments—are lacking. We leave the development of such datasets for future work.

**Limitations of our experiments**    In this work, we have run experiments under our proposed framework with two datasets and four state-of-the-art open-sourced LLMs. Despite the extensive experiments we have done, the findings with our experiments and framework are still specific to the LLMs and the particular task we have investigated, and should not considered universally applicable across other tasks and models. Nevertheless, the findings are important since they are new and complementary to existing literature. Further investigations could also examine how factors like fine-tuning methods, model scale, linguistic characteristics and training data distribution influence the abstractions in LLM representations.

## B    Broader Impact

Our work introduces a probing framework designed to identify the types of essential information about the underlying world that can be retrieved from LLM representations. Our experiments show that without this detailed analysis, discrepancies can arise between evidence and conclusions. Such discrepancies may lead to undue pessimism about LLMs' lack of world awareness or excessive optimism about their ability to develop general world representations. Consequently, we expect that our work could be potentially helpful to other researchers and LLM users, enabling them to draw more nuanced and comprehensive conclusions from analyzing LLMs regarding their ability to construct world representations. This in-depth understanding is crucial for developing responsible AI systems.

## C    Task Instructions and Input Examples in GRIPPER

Table 9 shows examples of each dataset variant in GRIPPER. Table 10 provides the full task instruction included in the prompt.

## D    Details and Examples of COOK Dataset

We generate the COOK dataset with TextWorld [8]. Concretely, we adopt the *cook* domain in TextWorld, where the agent navigates in a grid world with multiple rooms and manipulate the food ingredients to make a meal. Whereas TextWorld primarily focuses on evaluating RL with respect to react to the feedback from the environment, where the updated state information is included in the feedback, it makes the task of world modeling somehow trivial at a surface level. Therefore, we modify its setting to the one similar to GRIPPER, where only the initial and desired state are given in the prompt, and the models have to infer the intermediate world state induced by predicted or

ground-truth actions. Nonetheless, there are two main differences between COOK and GRIPPER. First, the containers (rooms) in COOK are placed in a $2 \times 3$ grid world, where the agent can move `south`, `north`, `east` or `west`, and therefore have a larger action space. In contrast, the agent move either `left` or `right` in GRIPPER. Second, COOK uses a different set of textual templates adopted from TextWorld, which are listed in Table 3. Table 11 shows the instruction for COOK and Table 12 provides an input example. COOK has the same size with GRIPPER.

| | GRIPPER | COOK |
|---|---|---|
| store | container *contains* object, 
 container *holds* object, 
 container *has* object, 
 container *is occupied by* object 
 container *is empty* 
 ... | container *contains* object, 
 In container, you can see object, 
 container *has* object *in it* 
 In container, you can find object, 
 container *has nothing* 
 ... |
| boxName | *A sequence of containers are ordered from left to right as follows:* Box A, Box B, ... | Room A *in the northwest connects east to* Room B *and south to* Room C. 
 Room A *links further east to* Room B *and south to* Room C. 
 Room A *connects south to* Room B. 
 Room A *leads east to* Room B. 
 Room A *connects east to* Room B. |
| grab | *Grab the* object *from nearby container* | *Take* object *from the room* |
| put | *Put* object *in the nearby container.* | *drop* object *in the room.* |
| move | *go* direction | *walk to* direction |
| Examples of $o$ (objects) | *book, car, paper, letter, game...* | *onion, tomato, potato, mushroom, carrot...* |

Table 3: Textual templates used in GRIPPER and COOK.

# E  Detailed Introduction of Abstract Predicates

We derive following abstract predicates using raw predicates, namely `boxName`, `store`, `agentLoc` and `held`:

- `nearby`$(o)$: the object $o$ is located within the container nearby to the agent. It is derived by incorporating `agentLoc` with `store`. Similarly, `distant`$(l, o)$, where $l$ denotes the relative distance to the agent's current location, can be derived. That means $l \in [-L+1, L-1]$, where $L$ denotes the number of containers. For instance, `distant`$(-1, o)$ being `True` indicates that the container one step to the left of the agent contains $o$.

- `subgoal`$(l, j)$: the agent needs to visit a series of containers to manipulate objects, and the $j$-th container to be visited is at the relative position $l$ to the agent's current location. Furthermore, for `subgoal`$(j, l)$, the label for $l$ is in the set $\{-L+1, \ldots, L-1\} \cup \{put, stop\}$, where `put` indicates the $j$-th subgoal is to put an object at the current nearby container, which is particularly useful for the first `subgoal`, as it allows for deterministic derivation of `nextObjDirect`. Also, `stop` implies that there are fewer than $j$ subgoals. `subgoal` is derived by comparing `distant`, `nearby` and `held` across the current and target situation.

- `nextObjDirect`$(v)$: the relative direction of the object to be grabbed or put at the next step, where $v$ could be `left` or `right` to the agent, in a nearby container, or with the agent. This is derived from the value of first `subgoal`, i.e., `subgoal`$(\cdot, 1)$.

- `nextObj`$(o)$: the object $o$ to be manipulated at the next step, if applicable. This is derived by comparing `nearby` and `held` between $u$ and $g$.

## F Adapting LLMs to REPLACE

We use Llama2-7b[5], Llama2-13b[6], Mistral-7b[7], Llama3-8b[8], Phi3-17b[9], Phi3-3.8b[10], and Pythia-70m[11] as the base models for experiments. Additionally, we train another Pythia-70m model from scratch on REPLACE datasets as a Transformer baseline.

For ICL, we randomly sample two demonstrations from the training set and append them to the prompt. For SFT, we train LoRA adapters [19] with 8-bit quantization on top of LLMs with the learning rate of $1e - 4$, the batch size of 8, the scaling factor of $\alpha = 64$, the dropout rate of 0.05, the rank of 32 for 1 epoch. To avoid significantly altering the LLM's representational traits, we use only a subset of 10k training samples for SFT. We use 8-bit quantization for decoding. During decoding, we parse the generated textual description $\hat{Y}$ of subsequent actions to its symbolic form with a rule-based parser. In our preliminary experiments, we have found using hard-crafted regular expression rules for the parser is good enough, since the the in-context demonstrations and the fine-tuning enforce well LLMs to generate in the same format as the data in `Gripper` and `Cook`.

## G Proof of World State Abstraction for REPLACE

Recall that the raw state $\mathcal{S}$ in REPLACE is factorized by `boxName`, `store`, `agentLoc` and `held`.

**Lemma G.1.** $\mathcal{S}_{\phi_w}$ *is a* world-irrelevant *abstraction of* $\mathcal{S}$ *if it is factorized by* `store`, `agentLoc` *and* `held`, *excluding* `boxName`.

*Proof.* Assume $\phi_w(s_1) = \phi_w(s_2)$. This implies that $\forall a \in \mathcal{A}$, $T_{\phi_w}(s'|s_1, a) = T_{\phi_w}(s'|s_2, a)$, given that `boxName` in $\mathcal{S}$ always remains unchanged after the execution of any action, we have:

$$
\begin{aligned}
&\forall x' \in \mathcal{S}_{\phi_w}, \\
&\sum_{s' \in \phi_w^{-1}(x')} T(s'|s_1, a) \\
&= \sum_{s' \in \phi_w^{-1}(x')} T\left([\phi_w(s'); \mathbf{b}_{s'}]|[\phi_w(s_1); \mathbf{b}_{s_1}], a\right) \\
&= \sum_{s' \in \phi_w^{-1}(x')} T_{\phi_w}\left(\phi_w(s')|\phi_w(s_1), a\right) \\
&= \sum_{s' \in \phi_w^{-1}(x')} T_{\phi_w}\left(\phi_w(s')|\phi_w(s_2), a\right) \\
&= \sum_{s' \in \phi_w^{-1}(x')} T\left([\phi_w(s'); \mathbf{b}_{s'}]|[\phi_w(s_2); \mathbf{b}_{s_2}], a\right) \\
&= \sum_{s' \in \phi_w^{-1}(x')} T(s'|s_2, a),
\end{aligned}
\tag{1}
$$

where $\mathbf{b}_{s'}$ denotes the assignment over `boxName` at $s'$. Similarly, it follows that $\phi_w(s_1) = \phi_w(s_2) \Rightarrow R(s_1, a) = R(s_2, a)$, since $R(\cdot, \cdot)$ is invariant with respect to changes in `boxName` within a state $s$. $\square$

---

[5] https://huggingface.co/meta-llama/Llama-2-7b-chat-hf (We accept their Meta license before requesting the model.)

[6] https://huggingface.co/meta-llama/Llama-2-13b-chat-hf (We accept their Meta license before requesting the model.)

[7] https://huggingface.co/mistralai/Mistral-7B-Instruct-v0.2 (licensed by Apache-2.0)

[8] https://huggingface.co/meta-llama/Meta-Llama-3-8B-Instruct (We accept their Meta license before requesting the model.)

[9] https://huggingface.co/microsoft/Phi-3-medium-4k-instruct (licensed by MIT)

[10] https://huggingface.co/microsoft/Phi-3-mini-4k-instruct (licensed by MIT)

[11] https://huggingface.co/EleutherAI/pythia-70m (licensed by Apache-2.0)

**Lemma G.2.** $S_{\phi_\pi}$ *is an* $\pi^*$-*irrelevant* *abstraction of* $\mathcal{S}$ *if* $S_{\phi_\pi}$ *is factorized by* `nextObj` *and* `nextObjDirect`.

*Proof.* To demonstrate that $S_{\phi_\pi}$ is a $\pi^*$-irrelevant abstraction, we show that the optimal action $a^*$ in any state can be uniquely identified using `nextObj` and `nextObjDirect`. This is formalized as follows:

$$a^* = \begin{cases} \texttt{grab(O}^*\texttt{)}, & \text{if } \texttt{nOD} = \texttt{container} \\ \texttt{put(O}^*\texttt{)}, & \text{if } \texttt{nOD} = \texttt{with-agent} \\ \texttt{move(left)}, & \text{if } \texttt{nOD} = \texttt{left} \\ \texttt{move(right)}, & \text{if } \texttt{nOD} = \texttt{right} \end{cases} \tag{2}$$

, where `nOD` denotes the value returned by `nextObjDirect` and $\texttt{O}^*$ denotes the object identified by `nextObj`. This ensures that for any two states $s_1$ and $s_2$ where $\phi_\pi(s_1) = \phi_\pi(s_2)$, it follows that $\pi^*(a|s_1) = \pi^*(a|s_2)$, $\qquad\qquad\square$

**Lemma G.3.** $S_{\phi_Q}$ *is an* $Q^*$-*irrelevant* *abstraction of* $\mathcal{S}$ *if* $S_{\phi_Q}$ *is factorized by* `agentLoc`, `held`$_u$, `nearby`, `subgoal` *and* `nextObj`.

*Proof.* We start by deterministically extracting a sequence of containers, to be visited, represented by their relative distances to the agent's current location $\mathbf{g}$. This sequence is denoted as $\mathbf{g} = [\mathbf{g}_1, ..., \mathbf{g}_j]$, where for each $i$, $\texttt{subgoal}(i, \mathbf{g}_j) = \texttt{True}$. `nextObjDirect` is inferred from $\mathbf{g}_1$. Consider the optimal action $a^*$ derived from `nextObj` and `nextObjDirect`, as discussed in Theorem G.2. The value of $Q^*(s, a^*)$ is given by:

$$-r \cdot \sum_{i=1}^{j} |\mathbf{g}_{i+1} - \mathbf{g}_i| - r \cdot j + r,$$

where the first term represents the cost of traveling between containers, the second term accounts for the cost of executing actions, and the third term is a positive reward for achieving the desired situation. It follows that $Q^*(s, a)$ for all actions can be measured as below:

$$Q^*(s, a) = \begin{cases} Q^*(s, a^*), & \text{if } a = a^* \\ Q^*(s, a^*) - r' - r, & \text{if } a \notin C(s) \\ Q^*(s, a^*) - r, & \text{else} \end{cases} \tag{3}$$

. where $C(\cdot)$ is the constraint function defined in Section 4, dependent on `nearby`$_u$, `held`$_u$ and `agentLoc`.

Consequently, $Q^*(s, a)$ can be precisely recovered conditioned on the predicates `agentLoc`, `held`$_u$, `nearby`, `subgoal`, and `nextObj`. This implies that $Q^*(\phi_Q(s_1), a) = Q^*(\phi_Q(s_2), a) \Rightarrow Q^*(s_1, a) = Q^*(s_2, a)$. $\qquad\qquad\square$

# H   Details about Design of Probing Tasks and Training of Probing Models

We detail the formulation and evaluation of the probing for each predicate as follows:

`boxName`: Given an LLM-built representation $h_t$, and a set of candidate container names $\mathbf{b}$, as well as the embedding $\varphi(b)$ for each $b \in \mathbf{b}$, the probing model $\mathcal{M}_\theta^i$ for $i$-position learns to select the ground-truth container name $b^*$ by maximizing the probability of $\texttt{boxName}(i, b^*)$ being `True`:

$$\mathcal{J}_\theta = -\log \frac{\exp\left(\mathcal{M}_\theta^i([h_t; \varphi(b^*)])\right)}{\sum_{b \in \mathbf{b}} \exp\left(\mathcal{M}_\theta^i([h_t; \varphi(b)])\right)}.$$

During testing, we include the container names in the same data instance as candidates and measure the F1-score using a macro average.

`agentLoc` and `nextObjDir`: Both of these predicates are formulated as multi-class classification tasks. Given $h_t$, the probing model learns to maximize the ground-truth label, i.e. $i^*$ for $\texttt{agentLoc}(i)$ and $v^*$ for $\texttt{nextObj}(v)$:

$$\mathcal{J}_\theta = -\log \frac{\exp\left(P_{\mathcal{M}_\theta}(i^*; h_t)\right)}{\sum_i \exp\left(P_{\mathcal{M}_\theta}(i; h_t)\right)}.$$

We adopt the F1-score as the evaluation metric.

`store`, `held` and `nearby`: These predicates may have multiple labels. For instance, a container may hold a number of objects. Therefore, we decompose a predicate into multiple independent decisions. For instance, we let the $i$-th probing model $\mathcal{M}_\theta$ for $\texttt{store}(i, \cdot)$ learns to perform pairwise binary classification for $\texttt{store}(i, o)$, where $o \in \mathbf{O}$:

$$\mathcal{J}_\theta = q(i, o) \cdot \log P_{\mathcal{M}_\theta^i}([h_t; \varphi(o)])$$
$$+ (1 - q(i, o)) \cdot \log \left( 1 - P_{\mathcal{M}_\theta^i}([h_t; \varphi(o)]) \right),$$

where $q(i, o) = [\![\texttt{store}(i, o)]\!]$. During testing, we combine all the predictions for $\texttt{store}(i, o)$ for all candidates of $o$. We use the exact-match score method to evaluate the predictions of $\texttt{store}(i, \cdot)$, $\texttt{held}(\cdot)$, $\texttt{nearby}(\cdot)$, and we calculate the average score as the final metric.

`nextObj`: We formulate and train the model in the same way as $\texttt{boxName}(i, \cdot)$. Again, we adopt the F1-score as the evaluation metric.

`subgoal`: As mentioned above, we train a separate probing model for the $j$-th subgoal. Each $j$-th subgoal is formulated, and the probing model is accordingly trained in the same way as `nextObjDirect` and `agentLoc`. We use the F1-score for evaluation.

When we compare probing performance across different predicates, we normalize all the metric score to the recovery rate, as described in Section 5.

We create probing data with GRIPPER and COOK as well as all variants of LLMs. Following [28], probing models are built upon two-layer fully-connected neural networks. Consistent with their observation, we have found that the probing models with only one layer have worse performance on some predicates. For instance, when tested with Llama2-13b , the probing performance for `nearby` and `subgoal` drops to $80.91$ and $55$, respectively, whereas performance for other predicates remains nearly unchanged (with variances of less than 1 score). On the other hand, using more layers (e.g. three layers) does not lead to improvement for any predicate. We explore two types of embedding methods $\varphi(\cdot)$ and select the one with the largest performance margin between using $[h_t, \varphi(e)]$ and using only $\varphi(e)$ on the validation set. As a result, we use word embedding for `boxName`, `store`, `held`, `nextObj`, and contextualized embedding for `nearby`. We explore multiple variants of $m$ for extracting $h_t^m$: the last layer, the last 6th layer, and the last 12th layer. Among these, the last 6th layer has been found to lead to the best overall performance on the validation set. We also explore different dimensions of neural probing models: 6600, 7600, 8600, and 9600, selecting 8600 based on performance on the validation set.

For the probing models takes as input the concatenation $[h_t, \varphi(e)]$, we uses a trick that transforms the it to $[h_t + \varphi(e); h_t - \varphi(e); h_t \cdot \varphi(e)]$ as the input of neural networks. In our preliminary experiment, we have found it leads to better probing performance.

All probing models were trained with the learning rate of $1e - 3$ and the batch size of 64 for 30 epochs. The main probing experiments were conducted three times, while the others were conducted once. All experiments were run using a single Nvidia A100 GPU, and each of them was completed within 24 hours.

# I  Complete Planning Experiment Results

Table 4 presents the planning performance of both variants of all LLMs for REPLACE.

# J  Complete Probing Experiment Results

Table 5 and Table 6 present the probing performance for different predicates across all variants of LLMs on GRIPPER and COOK, respectively. We conduct the probing experiment for each predicate with each LLM variant three times using different random seeds and report the mean value. We also report $\sigma$, the mean variance across different runs. Specifically, we average the variance across different LLM variants. The low variance observed indicates that our experimental findings are robust.

| | GRIPPER | | | COOK | | |
|---|---|---|---|---|---|---|
| Models | %Legal | %Succ | %Optim | %Legal | %Succ | %Optim |
| Llama2$_{\text{ICL}}^{7b}$ | 34.89 | 3.83 | 0.00 | 30.29 | 2.27 | 0.76 |
| Llama2$_{\text{ICL}}^{13b}$ | 62.41 | 3.07 | 0.19 | 35.41 | 1.70 | 0.36 |
| Mistral$_{\text{ICL}}$ | 41.12 | 2.12 | 0.65 | 44.41 | 1.95 | 0.22 |
| Llama3$_{\text{ICL}}$ | 62.19 | 10.36 | 2.17 | 39.11 | 2.26 | 0.09 |
| Llama2$_{\text{SFT}}^{7b}$ | 95.40 | 87.20 | 82.18 | 89.13 | 64.86 | 35.59 |
| Llama2$_{\text{SFT}}^{13b}$ | 96.54 | 88.30 | 84.02 | 95.71 | 85.11 | 69.17 |
| Mistral$_{\text{SFT}}$ | 97.07 | 92.15 | 87.36 | 95.04 | 85.91 | 61.47 |
| LLama3$_{\text{SFT}}$ | 96.25 | 89.14 | 74.32 | 95.36 | 86.61 | 61.21 |
| Phi3$_{\text{ICL}}^{17b}$ | 52.88 | 18.78 | 8.33 | — | — | — |

Table 4: Planning performance of LLMs on GRIPPER and COOK.

| Probing predicates | Abstraction type | Llama2$_{\text{SFT}}^{7b}$ | Llama2$_{\text{ICL}}^{7b}$ | Llama2$_{\text{SFT}}^{13b}$ | Llama2$_{\text{ICL}}^{13b}$ | Mistral$_{\text{SFT}}$ | Mistral$_{\text{ICL}}$ | Llama3$_{\text{SFT}}$ | Llama3$_{\text{ICL}}$ | Pythia-70m$_{\text{ICL}}$ | Phi3-3.8b$_{\text{ICL}}$ | Phi3-17b$_{\text{ICL}}$ | Phi3-17b$_{\text{SFT}}$ | Transformer | σ |
|---|---|---|---|---|---|---|---|---|---|---|---|---|---|---|---|
| boxName | Raw | 23.12 | 20.13 | 18.02 | 21.14 | 17.56 | 18.33 | 17.73 | 24.33 | 17.82 | 31.35 | 39.61 | 23.33 | 16.81 | 0.87 |
| store$_u$ | Raw, World-irrelevant | 7.86 | 1.98 | 7.84 | 3.52 | 8.61 | 8.09 | 7.82 | 8.16 | 6.15 | 7.01 | 8.34 | 8.69 | 9.97 | 0.29 |
| store$_g$ | Raw, World-irrelevant | 10.55 | 4.62 | 10.52 | 11.13 | 11.10 | 12.08 | 9.83 | 10.52 | 6.27 | 9.57 | 10.93 | 11.41 | 13.19 | 0.07 |
| held$_g$ | Raw, World-irrelevant | 6.13 | 0.99 | 6.31 | 6.60 | 6.06 | 5.50 | 6.42 | 5.61 | 0.56 | 5.72 | 6.38 | 6.16 | 6.06 | 0.21 |
| agentLoc | Raw, World-irrelevant, $Q^*$-irrelevant | 95.07 | 18.21 | 97.51 | 16.82 | 98.90 | 59.04 | 98.78 | 20.35 | 3.67 | 57.15 | 75.73 | 98.24 | 93.18 | 0.27 |
| held$_u$ | Raw, World-irrelevant, $Q^*$-irrelevant | 31.86 | 28.85 | 29.21 | 31.97 | 29.54 | 31.03 | 30.49 | 29.68 | 28.69 | 33.12 | 38.80 | 30.33 | 27.38 | 0.65 |
| nearby$_u$ | $Q^*$-irrelevant | 88.97 | 44.27 | 90.86 | 43.95 | 95.41 | 54.42 | 94.18 | 43.74 | 15.13 | 42.68 | 42.57 | 72.78 | 74.28 | 0.96 |
| subgoal | $Q^*$-irrelevant | 45.53 | 2.45 | 58.71 | 0.00 | 59.45 | 9.55 | 57.89 | 3.43 | 0.37 | 0.00 | 8.45 | 54.71 | 13.59 | 2.75 |
| nextObj | $Q^*$-irrelevant, $\pi^*$-irrelevant | 77.90 | 4.49 | 79.60 | 11.55 | 83.12 | 32.46 | 84.01 | 35.58 | 0.00 | 30.56 | 36.91 | 75.72 | 3.45 | 0.78 |
| nextObjDirect | $\pi^*$-irrelevant | 86.63 | 12.76 | 91.71 | 17.51 | 92.04 | 46.34 | 91.70 | 17.86 | 15.09 | 24.77 | 59.73 | 89.36 | 53.87 | 0.78 |

Table 5: Probing performance (recovery rate) of all predicates on GRIPPER.

| Probing predicates | Abstraction type | Llama2$_{\text{SFT}}^{7b}$ | Llama2$_{\text{ICL}}^{7b}$ | Llama2$_{\text{SFT}}^{13b}$ | Llama2$_{\text{ICL}}^{13b}$ | Mistral$_{\text{SFT}}$ | Mistral$_{\text{ICL}}$ | Llama3$_{\text{SFT}}$ | Llama3$_{\text{ICL}}$ | Transformer | σ |
|---|---|---|---|---|---|---|---|---|---|---|---|
| boxName | Raw | 12.52 | 13.53 | 12.21 | 15.57 | 10.64 | 14.95 | 11.97 | 12.88 | 24.49 | 0.30 |
| store$_u$ | Raw, World-irrelevant | 9.91 | 7.91 | 10.06 | 6.59 | 9.69 | 9.34 | 9.16 | 10.23 | 11.15 | 0.15 |
| store$_g$ | Raw, World-irrelevant | 12.58 | 8.96 | 12.58 | 8.23 | 11.95 | 9.32 | 11.02 | 7.85 | 13.62 | 0.21 |
| held$_g$ | Raw, World-irrelevant | 3.72 | 0.00 | 3.45 | 0.00 | 3.26 | 1.69 | 3.45 | 3.00 | 3.83 | 0.15 |
| agentLoc | Raw, World-irrelevant, $Q^*$-irrelevant | 99.30 | 15.00 | 99.13 | 16.53 | 99.69 | 34.72 | 99.61 | 14.84 | 63.83 | 0.38 |
| held$_u$ | Raw, World-irrelevant, $Q^*$-irrelevant | 42.19 | 40.16 | 37.33 | 34.88 | 38.47 | 41.38 | 40.90 | 40.78 | 26.36 | 0.21 |
| nearby$_u$ | $Q^*$-irrelevant | 92.76 | 44.94 | 90.03 | 30.50 | 96.09 | 34.95 | 96.56 | 45.27 | 41.72 | 1.36 |
| subgoal | $Q^*$-irrelevant | 47.07 | 6.57 | 54.53 | 9.95 | 62.92 | 7.63 | 64.22 | 2.87 | 29.84 | 0.66 |
| nextObj | $Q^*$-irrelevant, $\pi^*$-irrelevant | 77.48 | 0.00 | 74.51 | 0.00 | 69.20 | 0.00 | 76.58 | 0.00 | 0.56 | 0.26 |
| nextObjDirect | $\pi^*$-irrelevant | 87.52 | 6.77 | 88.39 | 9.59 | 91.87 | 18.01 | 90.21 | 12.22 | 57.41 | 0.68 |

Table 6: Probing performance (recovery rate) of all predicates on COOK.

# K    Probing Experiments with COLORGRIPPER

In the original setting of GRIPPER, the color $c$ of object $o$, denoted by $\text{color}(o,c)$, plays the role of distracting information that is totally irrelevant to task completion. We create another setting, COLORGRIPPER, where the color information is necessary to accomplish the task, and we compare the probing performance with respect to color information under these two settings. Specifically, COLORGRIPPER includes a new predicate $\text{color}_g$, the color of each object in the desired situations, and an action $\text{paint}(o,c)$, which paints the object $o$ in a nearby container to color $c$ and $c \in \{\text{red}, \text{blue}, \text{green}, \text{yellow}\}$. Also, we incorporate a new constraint that the new color must be different from the original one. As such, there is a new abstract state $\text{nearNotColor}(o,\ c)$, which classifies that the color of a nearby object $o$ is NOT $c$ and it pertains to $Q^*$-irrelevant abstraction. We fine-tune two Llama2-13b models on GRIPPER (Llama2$_{\text{org}}$) and COLORGRIPPER (Llama2$_{\text{color}}$) respectively, and probe the $\text{nearNotColor}$ and color from both fine-tuned LLMs. The results in Table 7 show that the probes fail to recover color, which do not belongs to goal-oriented abstraction under both settings, from Llama2$_{\text{org}}$ and Llama2$_{\text{color}}$. However, $\text{nearNotColor}$ can be probed from Llama2$_{\text{color}}$ with reasonable recovery rate. Again, this confirm our **finding 1** that LLM$_{\text{SFT}}$ tends to maintain a goal-oriented world abstraction during decoding.

# L    Probing from LLMs Fine-tuned with Suboptimal Action Sequences

Can the LLM representations become more general if we enforce greater random exploration within the environment? To answer this question, we synthesize another variant of GRIPPER, where ground-truth action sequences are sub-optimal. Concretely, they contain random legal moves that do not lead to the goal. We compare the average recovery rates (RR) for each abstraction from Llama3-8b models, fine-tuned on both the original and new datasets. The results are reported in Table 8. With the new dataset, the RR of goal-oriented abstractions decreases, which is expected as the model imitates a sub-optimal policy. However, the margin between goal-oriented and general abstractions is still substantial. More importantly, the RR of world-irrelevant abstraction does not increase at all, which implies that random exploration does not yield a more general world abstraction.

| Predicates | Abstraction Type | Llama2$_{org}$ | Llama2$_{color}$ |
|---|---|---|---|
| color | Raw | 0.00 | 1.92 |
| nearNotColor | $Q^*$-irrelevant | 0.00 | 37.70 |

Table 7: Comparative experiments of probing color information under GRIPPER and COLORGRIPPER.

| Dataset | Raw | World-irrelevant | $Q^*$-irrelevant | $\pi^*$-irrelevant |
|---|---|---|---|---|
| Original | 28.5 | 30.6 | 73.0 | 87.8 |
| Suboptimal | 25.5 | 27.4 | 63.3 | 72.1 |

Table 8: Comparative probing experiments with Llama3-8b fine-tuned on original and suboptimal actions on GRIPPER.

| | |
|---|---|
| *Base* | *A sequence of containers are ordered from left to right as follows: Crate A, Bucket A, Crate B, Basket A, Box A. You are at the Basket A.*
*Initial situation: Crate A contains a painting, a shoe. Bucket A contains a block, a disk. Crate B contains a glass. Basket A contains a rose. Box A contains a paper, a camera. You don't hold anything.*
*Desired situation: Crate A contains a painting, a shoe. Bucket A contains a block, a disk. Crate B contains a glass. Basket A is empty. Box A contains a camera. You hold rose, paper.* |
| *Lexical variants* | *A sequence of containers are ordered from left to right as follows: Box A, Basket A, Box B, Box C, Basket B. You are at the Box B.*
*Initial situation: Box A holds a small green petroleum, a big yellow gastropod. Basket A has nothing. Box B holds a small yellow sabre, a small red tricycle. Box C has a tiny green eggnog, a big yellow nametag, a small blue lantern, a big yellow biscotti. Basket B holds a tiny blue anaconda. You don't hold anything.*
*Desired situation: Box A is occupied by a petroleum, a gastropod. Basket A has nothing. Box B holds a sabre. Box C has a eggnog, a nametag, a lantern, a biscotti. Basket B has a anaconda, a tricycle. You don't hold anything.* |
| *Partial $\tau_s(g)$* | *A sequence of containers are ordered from left to right as follows: Box A, Box B, Box C, Basket A, Box D. You are at the Basket A.*
*Initial situation: Box A contains a jacket. Box B contains a boat, a paper, a ticket. Box C contains a bag. Basket A contains a engine. Box D contains a phone, a coffee, a ticket.*
*Desired situation: Box A contains a jacket. Box B contains a boat, a paper, a ticket. Box C contains a bag. Basket A contains a engine. Box D contains a phone.* |
| *Partial $[\tau_s(u); \tau_s(g)]$* | *A sequence of containers are ordered from left to right as follows: Bucket A, Basket A, Bucket B, Crate A, Bucket C. You are at the Crate A.*
*Initial situation: Bucket A contains a bomb, a ring, a medicine. Basket A contains a boat, a map, a bottle, a fish. Bucket B contains a dish, a fish. Crate A contains a plane, a stone. Bucket C contains a radio, a book, a map, a fish.*
*Desired situation: Bucket A contains a bomb, a ring. Basket A contains a boat, a bottle, a fish, a medicine. Bucket B contains a dish, a fish. Crate A contains a plane, a stone. Bucket C contains a radio, a book, a map, a fish.*
*Operations applied: You Move left for 2 steps.* |

Table 9: Examples of GRIPPER variants.

You are an intelligent agent in a fictional environment with a sequence of containers arranged in a straight line.
The environment involves movement, interaction with objects, and strategic planning to reach a particular state.
Structure of the Environment: There are containers arranged in a line from left to right. You are at one of them in the beginning.
Goals & Strategies: You start with an initial configuration of objects in containers. Your job is to reach the desired state, a different configuration, through a sequence of operations. If there is any object not mentioned in the desired state, that implies you should hold them in the end.
The legal operations include moving and manipulation(grab or put) of objects
1. Moving: You can only walk left or right. Each time you take a step, you'll be right next to the container on your left or right side.
2. manipulation: Inside some containers, there are objects. To grab an object or put one inside, you need to be at that container. If you see an object in a container far away, you can't grab it. You need to walk over to that container first.
Constraints: For each step, you can choose only one operation: either move or manipulate one object. You can hold multiple objects at once.

Table 10: The general task instruction in GRIPPER.

In TextWorld, You are an intelligent agent tasked with arranging food ingredients in a grid of rooms to match a specific desired configuration.
Structure of the TextWorld: TextWorld consists of a grid-based map with multiple rooms. You start in one of these rooms. Goals & Strategies: You start with an initial configuration of ingredients in rooms.
Your job is to reach the desired state, a different configuration, through a sequence of operations.
If there is any ingredient not mentioned in the desired state, that implies you should hold it in the end.
If you think the desired state is reached, terminate the process.
The legal operations include moving and manipulation(take or drop) of ingredients:
1. Moving: You can go north, south, west or east to enter adjoining rooms.
2. manipulation: Interact with ingredients inside the rooms by either taking or dropping them. You must be physically present in a room to interact with its contents. Constraints: For each step, you can choose only one operation: either move or manipulate one ingredient. You can hold multiple ingredients at once.

Table 11: The general task instruction in COOK.

You open the map of TextWorld. bedroom A in the northwest connects east to garden A and south to pantry A. garden A links further east to backyard A and south to garden B. backyard A connects south to bedroom B. pantry A leads east to garden B. garden B connects east to bedroom B. You are in backyard A.
Initial state: In bedroom A, you can see roasted sliced pecan, fried chopped mustard seed. garden A has grilled chopped nectarine, grilled chopped mango, raw chopped beef in it. backyard A has fried diced pork in it. pantry A contains raw sliced polenta, grilled diced rice, fried uncut tempeh. garden B has raw uncut mussels, grilled diced ham, roasted diced squid in it. bedroom B has grilled chopped nectarine, fried chopped mustard seed, roasted chopped walnut in it.
Desired state: bedroom A has nothing. In garden A, you can find nectarine, mango, beef, mustard seed. backyard A contains nothing. pantry A contains polenta, rice, tempeh. garden B contains mussels, ham, squid. In bedroom B, you can see nectarine, mustard seed, walnut.
Operations applied: You take pork from the room. You go west.

Table 12: An input example in COOK.

