# OpenReview forum: "Do LLMs Build World Representations? Probing Through the Lens of State Abstraction"
_NeurIPS.cc/2024/Conference — NeurIPS 2024 poster_

### Official Review · Reviewer_CXua · 2024-06-26

**Soundness:** 3
**Presentation:** 3
**Contribution:** 3
**Rating:** 7
**Confidence:** 4

**Summary:**

This work investigates what kind of abstractions LLMs use to encode the world, distinguishing goal-oriented abstractions (discarding world dynamics that are not necessary for achieving the goal) from world-general abstractions (including dynamics irrelevant for the goal). The authors note that prior work looking at LLM world models don't make this distinction, leading to conflicting results. For a text-based planning task, the authors probe LLMs doing the task in-context as well as fine-tuned, finding that from the latter goal-oriented abstractions can be recovered with a relatively higher accuracy than from the former.

**Strengths:**

- It's a neat idea and good contribution to formalise the types of abstractions LLMs can use to represent the world through state abstraction theory. I believe using this framework will make future work around these questions more interesting and grounded.
- The paper is well-written and easy to understand
- The authors design a synthetic task that is both easy to understand, has modular and distinct state abstractions, and is somewhat complex for LLMs to perform (requiring planning to perform optimally).

**Weaknesses:**

**LLMs can only do this task after task-specific fine-tuning, in which case it is unsurprising that the representations are goal-oriented. This doesn't mean that LLMs doing tasks out of the box (or in-context) well won't use world-general representations**

The main result is that LLMs fine-tuned for a task start forming more useful goal-oriented state abstraction representations. This is unsurprising, as when fine-tuning the LLM for a task you're essentially training it to discard irrelevant information and encode relevant information for the goal. I would expect world-general abstractions to only be present in a general model that can do a task well out of the box without task-specific fine-tuning. However, the result is presented in the paper like LLMs do not use world-general representations (e.g. from the abstrcct: "Our experiments reveal that LLM-built representations tend to favor goal-oriented abstractions during decoding,
prioritizing task completion over recovery of the world’s state and dynamics."). For this kind of claim to be sufficiently substantiated I would like to see the framework applied to a task that LLMs can do out of the box, or this task using LLMs that can do it in-context (e.g. larger models or better models). Alternatively, I'd like to see the text indicate that the claims are about task-specific (i..e. fine-tuned) LLMs.

**Missing baselines**

It's difficult to interpret the results without a simple baseline that learns a probe on top of a non-LLM trained to do this task. Now, we don't know whether the accuracy of around 20% for world-general predicates is because the LLM has picked up general knowledge from pretraining or because you can learn to represent these things from this task (either in-context or through fine-tuning.)

**Questions:**

**Questions**
- For section 6.1, it would be useful to know what percentage of random moves would be legal. It seems like the ICL performance of llama2 and mistral are very low / close to random?
- I would expect more world-general abstractions to be represented in earlier layers, and goal-oriented abstractions in the later layers. Did you look at the earlier layers for world-general predicate accuracy as well (first half)?

**Suggestions**
- I would caveat the abstract (and results mentioned elsewhere) sentence: "Our experiments reveal that LLM-built representations tend to favor goal-oriented abstractions during decoding, prioritizing task completion over recovery of the world’s state and dynamics.". You only find this for fine-tuned LLMs, and it might be entirely different for LLMs that can do a task out-of-the box.
- Should the predicate be subgoal(l=1, j) and subgoal(l>1, j) in line 270 - 271? It says in line 252 that l is the relative distance and j the j-th container to be visited.

**Limitations:**

Discussed and addressed in the appendix.

---

> ### Author Rebuttal · Authors · 2024-08-07
>
> Thank you for your time and constructive feedback. We address all of your concerns below.
>
> ### **Weakness 1: Results are unsurprising**
>
> > **(unsurprising finding)**  LLMs fine-tuned for a task start forming more useful goal-oriented state abstraction representations…  is unsurprising …
>
> * It's indeed surprising when viewed in the context of existing work$\color{red}{^1}$. We provide detailed elaboration in the [Global Rebuttal](https://openreview.net/forum?id=lzfzjYuWgY&noteId=i9JnaUoNpz), with other surprising findings and broader significance of our framework. We sincerely invite you to review it.
>
> > **(it's unsurprising because)** you're training it to discard irrelevant information and encode relevant information for the goal…
>
> * World dynamics are relevant to the goal since the optimal plan is derived based on predicting future states (illustrated in Fig.3).
> * On the other hand, LLMs-sft could have been abstracted away $Q^*$ abstraction since the $\pi^*$ abstraction alone is sufficient for predicting optimal actions. However, we successfully recovered $Q^*$ abstraction from LLM-sft representations. It reveals that LLMs-sft preserve the long-term impact of each possible action, despite being fine-tuned on next-token prediction only. To the best of our knowledge, this interesting finding is novel, and we believe it can inspire future research to conduct a more in-depth investigation.
>
> > **(experiments on a task that LLMs do well in context)** I would expect world-general abstractions to only be present in a general model that can do a task well out of the box without task-specific fine-tuning…
>
> * Given the space of possible tasks is extraordinarily huge, it's intractable to do an exhaustive search to identify a task that simultaneously satisfies **(1)** complex enough to differentiate the spaces of various abstractions, and **(2)** existing open-source LLMs can perfectly solve without fine-tuning. Even though we find such a task, it is hard to rule out the possibility of data contamination, especially considering that it still remains unknown if LLMs can solve a novel task that involves modeling the underlying world described by the text.
> * Due to these challenges, we instead start with a simple enough task. As explained in L179-184, the RePlace task closely resembles the [gripper problem](https://shorturl.at/j0wp5), one of the most basic planning tasks, where the transition is deterministic, the state can be derived without uncertainty, the goal must be achievable and the optimal plan can be searched in O(n). Therefore, we cannot make the task simpler while still having distinctive state space for different abstractions.
>
> > **(experiments with pre-trained LLMs)** …using LLMs that can do it in-context (e.g. larger models or better models)
>
> To fully address your concern, we conduct the same set of probing experiments with Phi3-17b, a brand-new SOTA LLM that achieves 19.94% success rate on GRIPPER with ICL only$\color{red}{^2}$. In response to your question that whether better LLMs are more likely to use general abstractions, we employ two smaller and weaker pre-trained LMs for comparative study. The new results indicate that (1) **goal-oriented abstractions are probed from pre-trained Phi3-17b with significantly higher recovery rate**, and (2) **more advanced pre-training leads to higher priority of encoding goal-oriented abstractions over a more general one**. Please refer to the Global Rebuttal for detailed results and in-depth analysis.
>
> > **(clarification needed)** I'd like to see the text indicate that the claims are about task-specific (i..e. fine-tuned) LLMs.
>
> Thank you for your suggestion! Due to space constraints, we summarize the findings for LLMs-sft/icl in one sentence in the abstract/Intro. In the experiments section (Sec 6.2), however, we have reported and analyzed the findings for LLMs-sft/icl separately (L320-321, L331). We will do the same thing for the abstract/Intro in the revised version to make it clear, which will be flexible given more space.
>
> ### **Weakness 2: learns a probe on top of a non-LLM**
>
> Thank you for your suggestion! To address your concern, we train two 6-layer decoder-only Transformers on our datasets, which achieves around 30% success rates. The average recovery rate of each abstraction is:
>
> |Datasets| Raw | World | $Q^*$ | $\pi^*$|
> | --- | --- |--- |--- |--- |
> |GRIPPER| 27.7 | 29.9  | 52.5  | 28.3|
> |COOK| 23.9 | 23.8  | 32.5 |28.7|
>
> Interestingly, the raw state and world-irrelevant abstractions can be probed with a much similar recovery rates as from LLMs-sft (\~30% for both). The main difference lies in goal-oriented abstractions, i.e. $Q^*$-irrelevant abstraction (\~73% from LLMs-sft) and $\pi^*$-irrelevant abstraction (\~83%). It suggests that pre-training mainly improves the encoding of goal-oriented abstractions.
>
> We will include this result in Fig. 5 in our paper. Thanks again for your valuable suggestion!
>
> ### **Questions**
> * Legal rate of random move: The legal rate of a random move is 21.65%. All LLMs-icl have significantly higher legal rates than this baseline.
> * Probing earlier layers: We treat the layer index as a hyperparameter (detailed in L620-L622) and use the last 6th layer based on the overall performance on the validation set. We have also experimented with using early layers. For instance, the average recovery rate (RR) of each abstraction from the 4th layer of Llama2-13b-sft is:
>
> |    Layer    | Raw | World | $Q^*$ | $\pi^*$|
> | --- | --- |--- |--- |--- |
> | 4th     | 13.7 |  13.0 | 14.7 | 7.02 |
>
> The RR of both the general- and goal-oriented abstractions are much lower.
>
> ### **Suggestions**
> * Thank you for the suggestion! We will summarize the findings for LLMs-icl/sft separately in the abstract/Intro to avoid confusion, which would be an easy fix.
> * Thank you for your question! `subgoal(l, j)` denotes that the container to be visited at $j$-th step is at relative distance $l$. We will clarify it in Line 270.

---

> ### Author Response · Authors · 2024-08-07
> **Footnotes and Reference list for Rebuttal**
>
> ### **Footnotes**
> $\color{red}{^1}$ We mainly focus on reporting the findings for LLMs-sft since it is a common interest [1, 2, 3, 4] in this field that if Transformers/LLMs trained on next-token predictions develop internal world models .
>
> $\color{red}{^2}$ While this result isn't very satisfactory, it is acceptable given that it's the best ICL performance we've seen with open LLMs. We didn't conduct probing experiments on COOK, because Phi3-17 still falls short of this dataset without fine-tuning (\~3% success rate). It's expected as COOK has a more complicated structure of containers, and larger space of actions.
>
> ### **Reference**
> [1] Li, Kenneth, et al. "Emergent World Representations: Exploring a Sequence Model Trained on a Synthetic Task." ICLR, 2023.
>
> [2] Li, Belinda Z., Maxwell Nye, and Jacob Andreas. "Implicit Representations of Meaning in Neural Language Models." ACL, 2021.
>
> [3] Hazineh, Dean S., Zechen Zhang, and Jeffery Chiu. "Linear Latent World Models in Simple Transformers: A Case Study on Othello-GPT." arXiv preprint arXiv:2310.07582 (2023).
>
> [4] Kim, Najoung, and Sebastian Schuster. "Entity Tracking in Language Models." ACL, 2023.

---

> ### Author Response · Authors · 2024-08-07
> **Thank you for your review; we're ready to answer any further questions**
>
> Dear reviewer,
>
> Thank you again for your review! We really appreciate your recognition of our methodologies, new resources, and presentation. Hopefully, our additional experiments in response to your comments and in-depth elaboration on the surprisal of our findings and the broader significance of our framework can address most of your concerns. If not, please don't hesitate to ask us any further questions. We are always ready to respond. We're very looking forward to an engaged and productive discussion!

---

> ### Comment · Reviewer_CXua · 2024-08-10
>
> Thanks for the detailed rebuttal, both personal and the general one. I will respond below.
>
> ## Using FT-ed models instead of base models
>
> The fact that world dynamics are still relevant for the goal and the fact that you recovered the $Q^*$ abstraction from LLM-sft representations is more convincing to me and alleviates some of my concerns. However, in the general rebuttal you claim your findings are more relevant in light of related work claiming that your finding:
>
>  *"undermines an increasingly widespread belief that an implicit world model can emerge from next-token prediction (NTP) training [1]. This belief is encouraged by recent studies demonstrating that world states can be recovered from Transformer representations [2,3]."*
>
> However, [3] explicitly controls for the effect on representations of pretraining and fine-tuning, from their paper section 4.2:
>
> *"To determine whether the advantage is conferred by LM pretraining or fine-tuning, we ablate either open-domain pretraining, in a -pretrain,+fine-tune ablation, or in-domain finetuning, in a +pretrain,-fine-tune ablation. [...] While both fine-tuning and pretraining contribute to the final probe accuracy, pretraining appears to play a much larger role: semantic state can be recovered well from models with no in-domain fine-tuning."*
>
> So even though for your task it might be the case that world-general representations can be useful, the results seemingly can't say much about whether or not pre-trained LLMs or next-word predictors learn world models.
>
> In your rebuttal, you say: *""Given the space of possible tasks is extraordinarily huge, it's intractable to do an exhaustive search to identify a task that simultaneously satisfies (1) complex enough to differentiate the spaces of various abstractions, and (2) existing open-source LLMs can perfectly solve without fine-tuning. Even though we find such a task, it is hard to rule out the possibility of data contamination, especially considering that it still remains unknown if LLMs can solve a novel task that involves modeling the underlying world described by the text.*
>
> Of course, I am not suggesting you do an exhaustive search over all possible tasks, as I don't even know what that would mean in practice, but it seems highly unlikely to me that there exists no task open source LLMs can perform in-context that affords both types of abstractions. Additionally, it's not necessary for the model to perform the task perfectly without fine-tuning. Finally, data contamination can be equally an issue in your fine-tuning setup, so I don't fully understand that argument.
>
> Finally, respectfully, I disagree that it's a good reason to not include the fact that your results are only on fine-tuned LLMs in the abstract due to space constraints. The line that is misleading is the following:
>
> "*"Our experiments reveal that LLM-built representations tend to favor goal-oriented abstractions during decoding, prioritizing task completion over recovery of the world’s state and dynamics.".*"
>
> Which can be changed to
>
> "Our experiments reveal that LLM-built representations when fine-tuned tend to favor goal-oriented abstractions during decoding, prioritizing task completion over recovery of the world’s state and dynamics.".
>
> ## Learning a probe
>
> Thanks for these additional insights. I believe it's important that these, together with the results on a pretrained model you have in the general rebuttal, are a part of the paper, as without it it remains hard to say much about which part contributes to what (pretraining vs finetuning).
>
> ## Summary
>
> To summarise, some of my concerns are alleviated, and I am still of the opinion that making the distinction between world-general and goal-oriented abstractions is a really cool contribution and makes for an interesting paper. The additional baselines and results on pretrained LLMs are interesting. This makes that I'll raise to score to a 5. The reason I did not raise it further just yet is as follows:
> I remain of the opinion that the results are somewhat unsurprising and do not necessarily go against prior findings. For example, you also find world-general abstractions in LLMs, just more goal-oriented ones. All in all I believe the paper requires quite a substantial rewrite to include the right baselines (non-LLMs), the experiments using a pretrained LLM from the general rebuttal, and to change the language about what exactly LLMs (when finetuned or not) do learn and how that relates to prior findings.

---

> > ### Author Response · Authors · 2024-08-11
> >
> > Dear reviewer,
> >
> > We've just saw your most updated response. We truly appreciate your active participation in the discussion phase, especially during what might also be a busy week for you. We're grateful to hear that we've addressed some of your concerns and that you find our findings interesting. This feedback boosts our confidence and excitement to share our work with a broader community. Thank you once again for your invaluable feedback; it has greatly enhanced the quality of our work! We will be sure to incorporate the promised changes in the revised version.

---

> > > ### Comment · Reviewer_CXua · 2024-08-12
> > >
> > > > "we can agree that SFT is an important type of NTP training, and our new experiments probing pre-trained LLMs address this concern."
> > >
> > > I do agree, and the main thing I was trying to get at was the way the results are presented in the paper. Indeed, with your additional experiments probing the pre-trained LLM the concerns are further alleviated, and the only thing that is left is the presentation.
> > >
> > > > "You might consider LLMs-icl results uninformative due to their poor performance on the task" and "Fine-tuning NTP doesn't result in a world model" and "pre-training on NTP doesn't inherently produce internal world models"
> > >
> > > Indeed, your results show that FT doesn't result in an internal world model for this task, and indeed the results on ICL are uninformative due to poor performance. Of course one can't such much about other tasks, and general claims about LLMs not having internal world models cannot be mode. This paper gives additional evidence to the full picture, but LLMs might very well be learning internal world models for other tasks. That is by itself not at all a weakness, such evidence is important, and I think your paper is a strong addition to the literature together with your supplemented results during the response phase. That's the reason why I increased my score, but not yet further than 5 because of the substantial rewrites I believe necessary.
> > >
> > > > "These results directly go against the claim from previous work that Transformers/LLMs trained on NTP develop internal world models. Moreover, we want to emphasize a meta-result that diverges from previous work: probing without considering world abstractions leads to biased evaluation and unnecessary conflicts, as reiterated throughout the paper and rebuttals."
> > >
> > > I agree, you're right, please excuse me for misrepresenting your results on this particular aspect in my previous response.
> > >
> > > >  "Incorporating the new results into the paper"
> > >
> > > This is my main remaining concern, and your details on how this will be done adresses my concerns.
> > >
> > > ## Summary
> > >
> > > Thanks for bearing with me, and thanks for responding to everything so thoughtfully. I am going to increase further to a 7. The way the authors engage in respond in this rebuttal phase make me relatively confident the proposed changes will actually be implemented. Both my weaknesses, about missing baselines (probe on top of non-LLM) and investigating pre-trained LLMs, are adequately addressed. That means the paper makes an important and interesting contribution of distinguishing goal-oriented and world-general abstractions when investigating world models, and present interesting findings that LLMs (both fine-tuned and general) do not encode world-general abstractions for their task. I do believe that to really show LLMs "in the wild" do not encode world-general abstractions it would need to be shown on more different and realistic types of tasks. Of course, there is always the risk of contamination, and that is a good reason to also show results on synthetic tasks, but for more general conclusions more tasks need to be considered. Nonetheless, the above contributions are sufficient for acceptance to me.

---

> > > > ### Author Response · Authors · 2024-08-12
> > > >
> > > > Dear Reviewer,
> > > >
> > > > We're thrilled to hear that most of your concerns have been addressed. We're truly grateful for your open-mindedness in considering our points and taking the time to review our responses thoroughly. To be honest, it's not often we encounter this level of engagement during the rebuttal process and we really enjoy our fruitful exchanges! We agree that assessing whether and how LLMs encode general world abstractions will be a long-term problem that demands continuous investigation. While our work may not provide a final answer, we hope it serves as a solid starting point. Rest assured, we'll make sure the promised changes are carefully implemented in the revised version!

---

> ### Author Response · Authors · 2024-08-10
> **Further Clarifications (1/3)**
>
> Thank you for your detailed feedback. We truly appreciate your engagement in further discussions and your acknowledgment that the facts we clarified in our rebuttal (the recovery of $Q^*$-abstractions and the relevance of world dynamics for the task) make our findings more surprising. It is also gratifying to know that our additional experiments echoing our original findings, provide deeper insights into world abstractions in both pre-trained and fine-tuned LLMs.
>
>
>
> We address every single one of your remaining concerns below.
>
> ---
>
>
>
> ### **Clarification about the finding based on SFT and its relation to a common belief shared by the community**
>
> In the Global Rebuttal, we stated that *"(Our $\color{blue}{\text{finding}}$ that SFT mainly enhances goal-oriented abstractions) undermines an increasingly widespread belief that **an implicit world model can emerge from next-token prediction (NTP) training** [1]"* and we cited two representative works [2,3] that encourage this belief. However, you raised concerns, arguing that our finding is insufficient because **[3]** *"explicitly controls for the effect on representations of pretraining and fine-tuning"* and they found *"While both fine-tuning and pretraining contribute to the final probe accuracy, **pretraining appears to play a much larger role**"*.
>
> Indeed, it doesn't negate the fact that our $\color{blue}{\text{finding}}$ presents a sufficient challenge to the belief discussed above, as we will elaborate below.
>
> (1) The next-token prediction (**NTP**) training mentioned in the belief is **not limited to pre-training alone**; the findings that encourage this belief go beyond just pre-training, including conventional supervised training for particular tasks (i.e., SFT or training from scratch). As you noted, even **[3]** has claimed that fine-tuning enhances world modeling. Moreover, the work [2] in this area that attracted wide attention, along with more recent follow-up work, e.g. [4], has probed from Transformer models trained on synthetic tasks with NTP. Therefore, our $\color{blue}{\text{finding}}$ directly addresses the belief since SFT on NTP is the de facto standard approach to adapt LLMs to a particular task.
>
>
> (2) Furthermore, the results you quoted from **[3]**, *"semantic state can be recovered well from models with no in-domain fine-tuning"*, don't undermine our finding that pre-trained LLMs struggle to maintain a general world abstraction (in RePlace), as a reanalysis of it [5] has already revealed that the high accuracy in **[3]** was based on trivial cases, and the entity status indeed cannot be recovered from pre-trained LMs with reasonable performance (please refer to section 2 in [5])$\color{red}{^1}$$\color{red}{^2}$.
>
> (3) Although we don't make such a claim in the paper, it's reasonable that one may expect some insights into the world modeling capabilities of pre-trained LLMs from the findings on LLMs-sft, as several recent studies suggest that it's very rare that fine-tuning alters the pre-trained capabilities [6,7,8]; rather, it tends to accentuate them. This may indicate that SFT could potentially enhance LLMs's prioritization of specific types of word abstractions that emerge during pre-training$\color{red}{^3}$. The findings from our new experiment with pre-trained LLMs, combined with the original ones, seem to align with this general hypothesis.
>
> With all that being said, we agree with you that **the $\color{blue}{\text{finding}}$ based on LLMs-sft should not be considered universally applicable to all LLM variants** (as we consciously warned in L495-498), including pre-trained LLMs. In response to your suggestion, we have conducted additional experiments, as detailed in the Global Rebuttal, to provide a more comprehensive assessment of the belief in question. Even if the findings from LLMs-sft do not fully address the world abstractions in pre-trained LLMs, hopefully, we can agree that **SFT is an important type of NTP training**, and **our new experiments probing pre-trained LLMs address this concern**.

---

> ### Author Response · Authors · 2024-08-10
> **Further Clarifications (2/3)**
>
> ### **Clarification about the conclusions from our experiments**
>
> >So even though for your task it might be the case that world-general representations can be useful, the results seemingly can't say much about whether or not pre-trained LLMs or next-word predictors learn world models.
>
> With great respect, we have to point out that this is NOT true.
>
> 1. In our original findings, world-irrelevant abstractions are mostly missing in both LLMs-sft and LLMs-icl, which are fine-tuned and pre-trained on NTP, respectively. While you might consider the LLMs-icl results uninformative due to their poor performance on the task, the LLMs-sft results clearly demonstrate that **fine-tuning on NTP doesn't result in an internal world model**.
> 2. Further experiments, prompted by your suggestion, on pre-trained LLMs and non-LLM transformers suggest that **pre-training on NTP doesn't inherently produce internal world models**—instead, more advanced pre-training leads to a higher priority for encoding goal-oriented abstractions.
>
> ---
>
>
> >  I remain of the opinion that **the results** … **do not necessarily go against prior findings**. For example, you also **find world-general abstractions in LLMs**, just more goal-oriented ones.
>
> This is NOT true, either.
>
> 1. Figure 6 in our paper and Plot B in the Global Rebuttal clearly show that **the predicate cannot be reliably recovered unless it pertains to the goal-oriented abstractions**.
> 2. Conversely, **the predicates uniquely tied to world-irrelevant abstraction**, namely `store_u`, `store_g`, and `held_g`, **are probed with a recovery rate of less than 12.5% across all LLM variants** (even lower than the one of `boxName`, which is unrelated to both task completion and world dynamics). Both types of NTP training—pre-training and fine-tuning—make little difference in this regard. It suggests that **LLMs tend to discard predicates that serve solely for preserving world dynamics**, the most critical ingredient of world models [9]. To be more concrete, LLM representations do not indicate how the world will change when an agent manipulates a certain object.
> 3. **These results directly go against the claim from previous work that Transformers/LLMs trained on NTP develop internal world models**.
> 4. Moreover, we want to emphasize a meta-result that diverges from previous work: probing without considering world abstractions leads to biased evaluation and unnecessary conflicts, as reiterated throughout the paper and rebuttals.
>
> ---
>
> ### **Why we don't work on a simpler task (such that LLMs can do well in context)**
>
> > Of course, I am not suggesting you do an exhaustive search over all possible tasks, as I don't even know what that would mean in practice, but it seems highly unlikely to me that there exists no task open source LLMs can perform in-context that affords both types of abstractions.
>
> We agree it's reasonable to expect a task that open-source LLMs can perform well in context, consisting of distinguishable world abstractions at various levels. However, finding such a task can be exhausting and tricky. Our task is already simple enough, as elaborated in our rebuttal and paper, so we didn't tweak the task setting until the LLM could achieve a reasonable performance. We were not implying that you suggested searching through all possible tasks. Apologize if there is any confusion.
>
> –--
>
> > Finally, data contamination can be equally an issue in your fine-tuning setup, so I don't fully understand that argument.
>
> Sorry for the confusion. By data contamination, we're referring to the possibility that the LLM might have been pre-trained on very similar or almost identical tasks. However, we're not suggesting that the LLM has encountered the exact same test samples during pre-training—this is a general concern across all setups. This possibility makes it unclear whether experimental results reflect SFT or pre-training effects. It's not an issue for fine-tuning setup, as we're fully aware that the LLMs are fine-tuned specifically for this task. If this still doesn’t seem like an issue to you, we can temporarily set it aside, as it doesn't affect the validity of our findings or how we've addressed your concern.
>
>
> Simply put, these three points in our rebuttal were to explain why we initially didn't conduct experiments on pre-trained LLMs. However, these were relatively minor, as we later identified a better model during rebuttal, Phi3 (released after the submission deadline), which achieves acceptable performance without fine-tuning. Therefore, the additional experiments in the Global Rebuttal primarily address your concerns regarding pre-trained LLMs.

---

> > ### Author Response · Authors · 2024-08-11
> > **Further Clarifications (3/3)**
> >
> > ### **Making the scope of findings clear in abstract/introduction**
> >
> >
> > > I disagree that it's a good reason to not include the fact that your results are only on fine-tuned LLMs in the abstract due to space constraints.
> >
> > To clarify the findings in our experiment section (**Finding 1**): (1) goal-oriented abstractions are much more effectively recovered from LLMs-sft representations than world-irrelevant abstractions, and (2) world-irrelevant abstractions are largely absent in both LLMs-icl and LLMs-sft representations. Thanks to your thoughtful reminder, we realize that we should have summarized the findings for LLM-sft/icl separately in the abstract and introduction. **As stated in our rebuttal, we will ensure this is clearly clarified in these two sections**. We are also open to addressing any remaining confusion.
> >
> > ---
> >
> > ### **Incorporating the new results into the paper**
> >
> > > I believe the paper requires quite a substantial rewrite to include the right baselines (non-LLMs) and the experiments using a pretrained LLM from the general rebuttal
> >
> >
> > In our humble opinion, these would be easy adjustments. First, the new results support our original findings, and second, incorporating them involves simply: (1) adding the new Plots A and B from the Global Rebuttal, presented in the same format as Figures 5 and 6 in our paper, along with the corresponding descriptions (points 1-3 in the Global Rebuttal); and (2) including the results with non-LLM baselines in Plot A and Figure 5, represented by one additional bar each.
> >
> >
> >
> > We sincerely hope the clarifications above encourage you to reassess the value of our work. We're eager to address any further concerns you may have. Thank you again for your time and dedication in the review process.
> >
> > ---
> >
> > ### **Footnotes**
> >
> > $\color{red}{^1}$ We are not intentionally downplaying the validity of [3]. We know it's one of the earliest works in this space, raising an important question and inspiring many follow-up works, including ours.
> >
> > $\color{red}{^2}$ We agree that it's important to distinguish the effects of pretraining from fine-tuning when drawing conclusions. Therefore, we compare probing results from LLMs-sft and LLMs-icl (L342-350, **Finding 2**) to investigate how SFT impacts the encoding of world abstractions. The new results (again, thanks to your suggestion) using pre-rained LLMs and non-LLM models help clarify the effects of pre-training, as discussed in the Global Rebuttal.
> >
> > $\color{red}{^3}$ To clarify, we're just explaining that this assumption is valid and inspired by existing work, and hence findings based on LLMs-sft could be informative and insightful. We're not suggesting that experimenting with LLMs-sft is sufficient for drawing conclusions for pre-trained LLMs.
> >
> >
> >
> > ### **Reference**
> >
> > [1] Kenneth Li, "Do Large Language Models learn world models or just surface statistics?", The Gradient, 2023.
> >
> > [2] Li, Kenneth, et al. "Emergent World Representations: Exploring a Sequence Model Trained on a Synthetic Task." ICLR, 2023.
> >
> > [3] Li, Belinda Z., Maxwell Nye, and Jacob Andreas. "Implicit Representations of Meaning in Neural Language Models." ACL, 2021.
> >
> > [4] Hazineh, Dean S., Zechen Zhang, and Jeffery Chiu. "Linear Latent World Models in Simple Transformers: A Case Study on Othello-GPT." arXiv preprint arXiv:2310.07582 (2023).
> >
> > [5] Kim, Najoung, and Sebastian Schuster. "Entity Tracking in Language Models." ACL, 2023.
> >
> > [6] Prakash, Nikhil, et al. "Fine-Tuning Enhances Existing Mechanisms: A Case Study on Entity Tracking." ICLR, 2024.
> >
> > [7] Panigrahi, Abhishek, et al. "Task-specific skill localization in fine-tuned language models." ICML, 2023.
> >
> > [8] Zhou, Chunting, et al. "Lima: Less is more for alignment." NeurIPS, 2024.
> >
> > [9] Ha, David, and Jürgen Schmidhuber. "Recurrent world models facilitate policy evolution." NeurIPS, 2018.

---

### Official Review · Reviewer_nz33 · 2024-07-04

**Soundness:** 3
**Presentation:** 3
**Contribution:** 3
**Rating:** 4
**Confidence:** 4

**Summary:**

This paper proposes a new framework for studying world state abstractions from LLM representations. The framework, based on state abstraction theory, focuses not on assessing whether a model has a single world representation but assessing different levels of possible abstractions. These are each roughly functions of states: world-irrelevant abstractions, q*-irrelevant abstractions, and pi*-irrelevant abstractions. A probe is used to assess whether an LLM's representation is encoding each abstraction. Specifically, the authors design a planning task called REPLACE (where each abstraction is known to the researcher but not to the model), and train a probe on a transformer's representation to predict the abstraction when prompted with a textual description corresponding to the abstraction.

In experiments, the authors prompt LLMs to perform the REPLACE task and assess which abstractions each LLM is encoding. They have four findings:
1. LLM representations reflect goal-oriented abstractions during decoding
2. Supervised-fine tuning increases the level of this goal-oriented abstraction
3. We don't see the other abstractions present during decoding
4. LLMs do not appear to build complete world representations.

**Strengths:**

I think the main strengths of the paper are as follows:

1. World abstractions: The idea of probing for different types of state abstractions is interesting. This is a new perspective for studying emergent world representations in LLMs, which have typically focused on all-or-nothing notions of world representations.

2. Clarity: The paper is mostly well-written and clearly structured. The authors do a good job of explaining their framework and describing the experimental methodology.

3. The planning task proposed in the paper (REPLACE) is an interesting task, and may be useful as a testbed for future research on analyzing world representations in language models.

**Weaknesses:**

While the framework proposed in the paper is interesting, the main weakness of the paper is that the experimental results are limited: REPLACE consists of two related planning task based on a limited set of containers and objects. Since the paper is focused on assessing world representations generally, there needs to be more evaluation settings and datasets. For example, the paper frequently refers to Othello as a testbed for assessing LLM world representations, but doesn't include any experiments involving Othello. The findings in the last section of the paper are based on insufficient evidence since they're only using the two related planning tasks that make up REPLACE. Do the same trends hold for game datasets? At the very least, more settings for REPLACE should be considered to provide more ablations of where each abstraction is recovered.

Similarly, the paper could benefit from more extensive model comparisons to make sure the takeaways are robust. While the authors do cover multiple LLMs, they're all from the Llama family or Mistral. Using models from different families (or at least the larger sizes of the Llama/Mistral models) would help make the findings more compelling.

Less importantly, there are a couple of writing and presentation suggestions that would improve the paper. The text in Figure 5 is difficult to read because it is small. Moreover, although Q* has a standard definition for anyone familiar with RL, it is not defined in the paper even though RL problems are defined.

**Questions:**

See questions above relating to robustness.

**Limitations:**

Yes

---

> ### Author Rebuttal · Authors · 2024-08-07
>
> Thank you for your time and detailed review. We address all of your concerns below.
>
> ### **Major Weakness: Running more experiments with more datasets/LLMs**
>
> > REPLACE … planning task based on **a limited set of containers and objects**.
>
> * We acknowledge that RePlace has predefined domains of containers and objects due to the synthetic nature of the datasets. However, **this synthetic setup is crucial for faithful probing experiments**.
> **(1)**  It allows us to manipulate and track the underlying world state for probing experiments.
> **(2)** We can deliberately balance the dataset based on various attributes, such as the order of containers and the initial location of agents and objects, to eliminate potential biases. However, it's hard to do the same thing for real-world datasets.
>
> * Due to the above reasons, almost **all existing work on probing world representations uses synthetic datasets**. [1,2] use game script datasets synthesized with the rule of the Othello game. [3,4] also employ synthesized text datasets based on limited sets of containers and objects, inspiring our datasets' design (mentioned in L175-176). However, we additionally have an agent exploring the environment and manipulating objects with some physical constraints to accomplish a goal, which makes the task more challenging and complex enough to differentiate the spaces of various abstractions. Furthermore, we have introduced different dataset variants to create a more diverse and realistic task setup (detailed in L227-234).
>
> * In addition, **our new task spans a broad domain** (elaborated in L179-182). It is closely related to the Gripper problem$\color{red}{^1}$ and the EntityTracking task [4,5], which is widely adopted in planning and NLP interpretability research. This task also involves tracking entities, aggregating multi-hop information, and recognizing textual entailment, essential skills for many real-world NLP tasks.
>
> > Since the paper is focused on assessing world representations generally, there needs to be **more evaluation settings and datasets**.
>
> * Our framework, the central contribution of this work, is generic and can be seamlessly applied to other LLMs and tasks wherever the abstract state spaces are distinguishable. However, we are not aiming to reach a conclusion that applies universally to all LLMs and tasks (discussed in L494-498). This is also impossible, given that there is even a lack of available testbeds.
> * We didn't use other datasets employed by existing work as **they don't meet the necessary criteria for faithful and feasible probing** (explained in L179-184, L488-493): (1) processing text data; (2) the abstractions across various levels are distinguishable; (3) the world state can be easily manipulated and tracked. Our new task/datasets fill this gap.
>
> > … doesn't include any experiments involving Othello
>
> * In the paper (L166-170), we explain **why the Othello dataset is not a faithful testbed**: it's not complex enough to differentiate abstract states at various levels.
> * Nonetheless, the Othello dataset was originally used to probe a small-scale Transformer trained on the game script data. But our work focuses on LLMs, and game script is very different from natural text that LLMs are trained on.
>
> > … more settings for REPLACE should be considered to provide more **ablations** …
>
> * **We have done such ablation**. Due to the limit of space, we highlight the conclusion in the main body of the paper (L337-L340) and provide more details in Appendix K. To sum up, we have created a new setting where an originally goal-unrelated predicate (e.g. color info) belongs to goal-oriented abstractions. As such, the recovery rate of color information has significantly increased from zero. This further consolidates our finding that SFT mainly enhances goal-oriented abstractions.
> * During the rebuttal period, we have done another ablation to answer Reviewer 4mUW's question, which shows that LLMs fine-tuned with sub-optimal demonstrations still prioritize goal-oriented abstractions, echoing our main finding. For the sake of not repeating, we sincerely invite you to refer to the rebuttal to Reviewer 4mUW for more details and results of this ablation study$\color{red}{^2}$.
>
> > … could benefit from **more extensive model comparisons** … all from the Llama family or Mistral…
>
> * We have already adopted four LLMs for experiments, each of which is experimented with both ICL and SFT, while existing work like [1,2] uses only one and [3] uses two. Even though some of the experimented LLMs are from the same family, they are at different scales. Llama2 and Llama3 have different tokenizers trained on different data and post-training methods (DPO, etc.).
> * To address your concern fully, we adopt another SOTA LLM, phi3-17b, and conduct new probing experiments. The results show that this SOTA LLM, whether or not it has been fine-tuned, tends to maintain a more goal-oriented abstraction of the underlying world. It is totally consistent with our original findings. Please refer to the [Global Rebuttal](https://openreview.net/forum?id=lzfzjYuWgY&noteId=i9JnaUoNpz) for more details and analysis.
> * Consequently, we end up experimenting with **12** LLM variants (5 SFT + 4 ICL + 3 pre-trained), which, to the best of our knowledge, is **the most extensive comparison**$\color{red}{^3}$ so far in the area of probing world representations.
>
> ### **Minor Weakness: text in Figure 5 too small & formal definition of $Q^\*$**
>
> Thank you for your suggestions! We will increase the font size in Fig.5 and quote the formal definition of $Q^*$ in the revised version. They would be very easy fixes.

---

> > ### Comment · Reviewer_nz33 · 2024-08-09
> >
> > Thank you for the rebuttal. While I appreciate the additional experiments and clarifications, I still have my original concerns about whether these conclusions can be made from just REPLACE. Therefore I will keep my original score.

---

> ### Author Response · Authors · 2024-08-07
> **Footnotes and Reference List for rebuttal**
>
> ### **Footnotes**
>
> $\color{red}{^1}$ Gripper problem is a classic AI/planning task featured in the first ICAPS's competition [6].
>
> $\color{red}{^2}$ In case you can't find it quickly, we've copied the details here. We synthesize another variant of GRIPPER, where ground-truth action sequences are sub-optimal, including some random legal moves that do not lead to the goal. We compare the average recovery rates for each abstraction from Llama3 models, fine-tuned on both the original and new datasets:
>
> |    Models    | Raw | World | $Q^*$ | $\pi^*$|
> | --- | --- |--- |--- |--- |
> | Llama3-sft (original, reported in paper)  | 28.5 | 30.6  | 73.0  |  87.8  |
> | Llama3-sft (new)  | 25.5 |27.4 | 63.3| 72.1 |
>
>
> The results show that the recovery rate of goal-oriented abstractions decreases, which is expected as the models imitate a sub-optimal policy. However, the margin between goal-oriented and general abstractions is still substantial. More importantly, the recovery rate of world-irrelevant abstraction doesn't increase at all, implying that training with non-optimal ground truths doesn't lead to a more general world abstraction. These new results echo our main findings, and we'll add them to the revised version.
>
> $\color{red}{^3}$ While existing work focuses on **one or two** predicates, our probing experiments involve **ten** different predicates, as our framework requires a comprehensive assessment of various types of abstractions. Also, the main results reported in the paper are averaged across multiple runs. In total, **we completed \~500 runs of training and testing probes across all LLMs and predicates**.
>
>
> ### **Reference**
>
> [1] Li, Kenneth, et al. "Emergent World Representations: Exploring a Sequence Model Trained on a Synthetic Task." ICLR, 2023.
>
> [2] Hazineh, Dean S., Zechen Zhang, and Jeffery Chiu. "Linear Latent World Models in Simple Transformers: A Case Study on Othello-GPT." arXiv preprint arXiv:2310.07582 (2023).
>
> [3] Li, Belinda Z., Maxwell Nye, and Jacob Andreas. "Implicit Representations of Meaning in Neural Language Models." ACL, 2021.
>
> [4] Kim, Najoung, and Sebastian Schuster. "Entity Tracking in Language Models." ACL, 2023.
>
> [5] Prakash, Nikhil, et al. "Fine-Tuning Enhances Existing Mechanisms: A Case Study on Entity Tracking." ICLR, 2024.
>
> [6] McDermott, D.M., 2000. The 1998 AI planning systems competition. AI magazine, 21(2), pp.35-35.

---

> ### Author Response · Authors · 2024-08-07
> **Thank you for your review; we're ready to answer any further questions**
>
> Dear reviewer,
>
> Thank you again for your review! We really appreciate your recognition of our methodologies, new resources, and presentation. We hope our rebuttal clearly explains why our new task representative and the existing testbeds are not ideal for faithful probing. Along with the two ablation studies and additional experiments on new families of LLMs, we hope that your suggestion for more extensive experiments has been adequately addressed. If there are any specific setups you believe might challenge our claims/findings that we haven't covered, please let us know. We'd be glad to accommodate them if feasible. We're very looking forward to an engaged and productive discussion!

---

> ### Author Response · Authors · 2024-08-09
>
> Thank you for your feedback. We regret that your concern about experiments on other tasks remains, although we believe we have thoroughly addressed the need for more extensive model comparisons in our rebuttal. If possible, we would like to respectfully reiterate several points for your kind consideration.
>
>  (1) Existing testbeds are NOT available for feasible and faithful probing;
>
>  (2) Due to (1), we develop a new one, creating TWO datasets in one effort;
>
>  (3) We have conducted two ablations with different task variants, fully addressing your request for "**more settings for ablations**";
>
>  (4) We aren't aware of any work analyzing LLMs/neural models that proposes two or more tasks simultaneously.

---

### Official Review · Reviewer_9vdN · 2024-07-09

**Soundness:** 3
**Presentation:** 3
**Contribution:** 2
**Rating:** 6
**Confidence:** 3

**Summary:**

This paper investigates whether different levels of world abstractions can be decoded from LLM representations. The study is performed using a synthetic dataset of simple planning problems involving moving objects between containers. The study takes inspiration from RL to define different levels of abstraction, from specific goal-oriented states to general world-related states. The LLMs (Llama and Mistral models) are initially adapted to the task through in-context learning and fine-tuning. The problem is encoded in natural language and the hidden states are extracted and used to train probing classifiers to determine whether each level of abstraction can be recovered. The results show that the recovery of goal-related states is much more successful than general world abstractions.

**Strengths:**

1. The presentation is rigorous and comprehensive.

2. The synthetic dataset is very good - simple but to the point. Upon release, it could be helpful for various other research papers.

3. The experiment performed using the dataset is clear and well-designed.

4. The question of whether LLMs build state abstractions is very interesting and currently an open question. Progress in this area would lead to more interpretable models.

**Weaknesses:**

Major comments:

1. Given that the models are first fine-tuned to the task, I wonder whether the results are not a self-fulfilling prophecy of the setup. If full problems and solutions were presented during adaptation, then it seems expected that the model would produce representations that are relevant to this task - the paper result would be, in a sense, a measure of the LLM having adapted to the fine-tuning, as opposed to reflecting the general inner workings of an LLM faced with a realistic or novel problem. A more convincing experiment would involve no task adaptation.

2. It is not clear how the hidden states were selected for probing. Only the "last hidden states" were selected (section 3.2). This again seems to invite the paper's results - it seems like economically preserving only relevant representations in the last layers is a measure of the LLM adaptation to the task. How about early layers of the LLM - can general abstractions be recovered there? Additionally, can any abstractions be recovered at all before adaptation?

Minor comments:

3. Some of the figures lack clarity. Figures 5-6 have unreadably small fonts, tables have barely readable fonts. "Best viewed in color" is a useful warning, but better colour selection would be ideal.

4. Typos: L326 "the most coarsest". Additionally, expressions like L318 "LLMs prefer maintaining a goal-oriented world abstraction over a more general one" are excessive LLM personification.

**Questions:**

See above.

**Limitations:**

Adequately discussed, but only in the Appendix, not in the main text.

---

> ### Author Rebuttal · Authors · 2024-08-07
>
> Thank you for your thoughtful and encouraging feedback. We address all of your concerns below.
>
> ### **Major Weakness 1: Findings are expected**
>
> > **(Findings unsurprising)** If full problems and solutions were presented during adaptation, then it seems expected that the model would produce representations that are relevant to this task
>
> * World dynamics are also relevant to the task as the optimal plan is derived based on the prediction of future states (illustrated in Figure 3), but they are almost absent from LLM representations.
> * On the other hand, LLMs-sft could have omitted $Q^*$ abstraction since the $\pi^*$ abstraction alone is sufficient for predicting optimal actions. However, we successfully recovered $Q^*$ abstraction from LLM-sft representations. It reveals that LLMs-sft preserve the long-term impact of each possible action, despite being fine-tuned on next-token prediction only. To the best of our knowledge, this exciting finding is novel and can inspire future research to conduct more in-depth analysis.
> * In the [Global Rebuttal](https://openreview.net/forum?id=lzfzjYuWgY&noteId=i9JnaUoNpz), we further elaborate on the broader significance of our framework and explain why our findings are surprising. We sincerely invite you to review it.
>
>
> > **(Experiments with pre-trained LLMs)** A more convincing experiment would involve no task adaptation.
>
> To completely address your concern, we conduct new probing experiments on top of Phi3-17b, a brand-new SOTA LLM that achieves 19.94% success rate on GRIPPER without any fine-tuning. From its representations, goal-oriented abstractions can be probed with a much higher recovery rate than the world-irrelevant abstraction. We further include two smaller and weaker pre-trained LMs for comparison. The results suggest that more advanced pre-training leads to a higher tendency to prioritize goal-oriented abstractions over a more general one. Please refer to the [Global Rebuttal](https://openreview.net/forum?id=lzfzjYuWgY&noteId=i9JnaUoNpz) for detailed results and in-depth analysis.
>
> ### **Major Weakness 2: Clarity issue**
>
> > **not clear how the hidden states were selected** … How about early layers of the LLM…
>
> * To clarify, the *last hidden states* refers to the hidden states at the last step before decoding, following the common practice of existing work. We treat the layer index as a hyperparameter (detailed in L620-L622) and use the last 6th layer for all experiments.
> * We have also experimented with using early layers. For instance, the average recovery rate (RR) of different abstractions from the 4th layer of Llama2-13b-sft is as follows:
> |    Layer    | Raw | World | $Q^*$ | $\pi^*$|
> | --- | --- |--- |--- |--- |
> | 4th     | 13.7 |  13.0 | 14.7 | 7.02 |
> | Last 6th | 28.2  |30.2 | 71.1| 85.6 |
>
> The RR for both the general- and goal-oriented abstractions significantly decreases. Therefore, the world abstraction cannot be recovered using earlier layers.
>
>
> ### **Minor Weaknesses**
>
> * **small fonts in the table and better color selection**: Thank you for your feedback! We will enlarge the font in the figure and use deepened color in the revised version.
> * **Typos and expression**: Thank you for your careful reading! We will fix the typos and grammar in the revised version. They would be very easy fixes.

---

> ### Author Response · Authors · 2024-08-07
> **Thank you for your review; we're ready to answer any further questions**
>
> Dear reviewer,
>
> Thank you once again for your review! We are truly grateful for your appreciation. We hope our rebuttal clearly explains why our findings are surprising and important, and that our additional experiments with pre-trained LLMs, along with our clarification about the selection of hidden states, address all of your concerns. Please don't hesitate to ask any further questions. We are always ready to respond, and we're very looking forward to an engaged and productive discussion!

---

> ### Author Response · Authors · 2024-08-11
>
> Dear Reviewer,
>
> We appreciate the effort you've put into reviewing our paper and offering really helpful feedback. We're wondering if you've had a chance to go through our responses. As the discussion period is nearing its close, we want to ensure that any remaining questions or concerns are fully addressed. If you feel that the main concerns (suprisal of findings & the selection of hidden states) have been satisfactorily resolved, could you please consider increasing the score to reflect that? We would be truly grateful.

---

### Official Review · Reviewer_4mUW · 2024-07-11

**Soundness:** 4
**Presentation:** 4
**Contribution:** 3
**Rating:** 7
**Confidence:** 4

**Summary:**

This work makes the observation that prior work in probing LLM for planning tasks comes to different conclusions on whether there are internal state abstractions in LLM hidden layers. This work hypothesizes that the disagreement comes from that these works are probing LLM with different tasks, which may necessitate LLM to learn abstraction at different granularities best suited for next-token prediction. Hence probing without controlling the abstraction level might give rise to different recovery rates of abstractions from hidden layers for different tasks. To support this claim, the paper borrows from RL literature the concept of world-/Q*-/$\pi^*$-irrelevant abstractions to designate different abstraction levels and aims to show whether they can recover these abstractions from hidden layers of Llama and Mistral models (both via in-context prompting and LoRA fine-tuning)

**Strengths:**

1. The paper is really well-motivated and written. The problem is a varying interesting one and the observation is timely for the field.
2. It's quite novel to break down abstractions through the lens of q value function and policy to capture aspects that are relevant for planning.
3. The design of the abstraction together with the RePlace experiment is rigorous and well-thought-out.
4. Figures are super helpful to ground analysis in concrete examples.

**Weaknesses:**

The biggest weakness I find is that the experiment may not be sufficient to support their central hypothesis. While it is necessary to show high recovery rate of Q*-/$\pi^*$-irrelevant abstractions for RePlace tasks, it is not sufficient to substantiate the claim that LLM will learn different abstractions given different tasks. RePlace tasks (both Gripper and Cook) is just one category of tasks whose solution requires forming abstractions such as Q value. Necessary additional experiments should show figure 6 on other tasks where 1. world-irrelevant features are important and 2. either Q abstractions or $\pi$ abstractions are important but not both to solve the task. If figure 6 shows different recovery rates for these other categories of tasks, there is sufficient evidence to support your claim.

**Questions:**

A lot of the authors' findings (e.g. finding 2 "Supervised fine-tuning mainly enhances a goal-oriented world abstraction") are not necessarily true because they only fine-tune LLM on RePlace tasks that are inherently goal-oriented. Finetuning with other categories of tasks that are not goal-oriented may not lead to goal-oriented world abstractions. For example, if you modify your RePlace domains to generate a dataset of non-optimal plans (potential back-and-forth legal moves but no shortest paths to goals), fine-tuning with this dataset might not lead to purely goal-oriented abstractions.

**Limitations:**

The authors make the remark that "one cannot determine whether the success of probing stems from the LLM’s preference for learning a general world model or from the necessity to recover the world state while learning the optimal policy" in the critique of prior work using the Othello game. However, the proposed probing method might not prevent supervised fine-tuning on a particular task from washing out LLM's innate preference either. In other words, the probing results only reflect abstractions suited to solve the tasks they fine-tune on. Perhaps one can only use ICL results to probe LLM's innate preference for abstraction learning, and to that end, one might need to come up with tasks that LLM are better at solving zero-shot.

---

> ### Author Rebuttal · Authors · 2024-07-31
>
> Thank you for your time and review. We appreciate the opportunity to clarify two critical aspects that may have been misinterpreted: our claims/hypothesis and the concept of goal-oriented abstractions.
>
> ### **Weakness**
>
> > This work hypothesizes that the **disagreement** comes from that these works are probing LLM with different tasks, which may necessitate LLM to learn abstraction at different granularities best suited for next-token prediction (from **summary**) …  the claim that LLM will learn different abstractions given different tasks (from **weakness**) …
>
> **Indeed, these are NOT our hypotheses and claims, and NONE of our contributions (listed in L74-83) & findings (Sec 6.2) is about LLMs learning different abstractions for different tasks.**
>
> We clarify our main claims & proofs below.
>
> * The disagreement among existing works is likely due to **(1)** they probe **only the raw world state** on different tasks (L30-31), and **(2)** the raw world state might equal to or overlap with varying levels of abstraction depending on the task (L34-L38).  It's, therefore, plausible that LLMs consistently preserve certain types of abstractions, but previous studies fail to capture this pattern.
>
> * **More critically**, probing WITHOUT the notion of world abstractions leads to biased evaluations, which may underestimate LLMs' world modeling when there is no overlap between the raw state and any abstraction or overestimate when the raw state aligns with goal-oriented abstractions (illustrated in L49-54, L164-170). Our framework directly addresses this limitation by probing different levels of abstractions$\color{red}{^1}$.
>
> * Within our framework, we primarily investigate **which types of world abstractions are encoded in LLM representations** (mentioned throughout the paper,  L71-73, 77-79, 145, 243-245). Sec 6.2 lists all findings. Moreover, **the results prove that probing only the raw state causes biased evaluations and unnecessary disagreement** (emphasized in L374-379, also elaborated in the Global Rebuttal).
>
> Due to the **lack of faithful testbeds** (explained in L162-170)$\color{red}{^2}$, we CANNOT further test if LLMs preserve different types of abstractions for other tasks$\color{red}{^3}$ .
>
>
> > **RePlace tasks … is just one category of tasks whose solution requires forming abstractions such as Q value**. Necessary additional experiments should show figure 6 on other tasks where 1. **world-irrelevant features are important** and 2. **either Q abstractions or $\pi$ abstractions are important but not both to solve the task**.
>
> This question doesn't hold up because
> (1) **The world-irrelevant abstraction is indeed important in RePlace** as it's necessary for predicting future states, from which the optimal plan is derived (illustrated in Fig.3).
> (2) **A goal-oriented abstraction unimportant for a task doesn't exist**$\color{red}{^4}$. State abstractions are task-specific and derived from the task's characteristics (i.e., transition dynamics and reward function). A state abstraction is a function that maps raw states to a smaller space, preserves the interested statistics (such as $Q^*$ and $\pi^*$), and omits other irrelevant information. Therefore, **if a predicate is not important** (no impact on the $Q^*$ or $\pi^*$) for a task, **it should be excluded from the goal-oriented abstractions**. Please refer to Sec 3.1 for formal definition and explanation of state abstractions as well as pointers to relevant literature.
>
> ### **Questions**
>
> >  Finetuning with other categories of **tasks that are not goal-oriented** may not lead to goal-oriented world abstractions.
>
> To clarify, we interpret "tasks that are not goal-oriented" as fine-tuning with sub-optimal ground truths, given that any RL task has a goal that maximizes the accumulated reward. We address this concern below.
>
> > For example, if you **modify your RePlace domains to generate a dataset of non-optimal plans** (potential back-and-forth legal moves but no shortest paths to goals), fine-tuning with this dataset might not lead to purely goal-oriented abstractions.
>
> This is an interesting question! To answer it, we synthesize another variant of GRIPPER, where ground-truth action sequences are sub-optimal (as you suggested, with some random legal moves that do not lead to the goal). We compare the average recovery rates (RR) for each abstraction from Llama3 models, fine-tuned on both the original and new datasets:
>
> |    Dataset    | Raw | World | $Q^*$ | $\pi^*$|
> | --- | --- |--- |--- |--- |
> | original (reported in paper)  | 28.5 | 30.6  | 73.0  |  87.8  |
> | new  | 25.5 |27.4 | 63.3| 72.1 |
>
>
> On the new dataset, the RR of goal-oriented abstractions decreases, which is expected as the model imitates sub-optimal actions. However, the margin between goal-oriented and general abstractions is still substantial. More importantly, the RR of world-irrelevant abstraction doesn't increase at all, which proves that training with non-optimal ground truths doesn't lead to a more general world abstraction. These new results echo our main findings, and we'll add them to the revised version.
>
> ### **Limitations**
>
> > However, the proposed probing method might not prevent supervised fine-tuning on a particular task from washing out LLM's innate preference either…
>
> It's completely flexible for our framework to verify this assumption by directly probing pre-trained LLMs. To demonstrate that, we conduct the same probing experiments with three different pre-trained LMs. The results suggest that more advanced pre-training increases the tendency to prioritize goal-oriented abstractions over more general ones, and that SFT further enhances this trend. Please refer to the [Global Rebuttal](https://openreview.net/forum?id=lzfzjYuWgY&noteId=i9JnaUoNpz) for detailed results and in-depth analysis.
>
> Moreover, we thoroughly discuss all aspects of our work's limitations in Appendix A, which includes the scope of our experimental findings.

---

> ### Author Response · Authors · 2024-08-07
> **Footnotes for rebuttal**
>
> $\color{red}{^1}$ To clarify, we don't need to use different tasks to probe different levels of abstraction. Instead, we derive the abstraction functions, such as world-irrelevant abstraction and $Q^*$-irrelevant abstraction, for the same task and then probe the abstract state at various levels from LLM representations. All we need is to carefully select a task where the spaces of abstract states at different levels differ (highlighted in L162-166). This generic framework is introduced in Section 3, and we demonstrate how to apply it to our new task, RePlace, in Section 5.
>
> $\color{red}{^2}$ In particular, there are three necessary criteria for faithful and feasible probing (explained in L179-184, L488-493): (1) processing text data; (2) the abstractions across various levels are distinguishable; (3) the world state can be easily manipulated and tracked.
>
> $\color{red}{^3}$ However, our findings can explain and resolve conflicts among existing works. Specifically, the contradiction may stem from various degrees of alignment between the raw states and goal-oriented abstractions across tasks. For instance, [1] successfully probes the Othello board state, which is already the goal-oriented abstraction for predicting legal moves (the task Transformers are trained on), whereas [2] struggles to probe the status of entities, as they have little overlap with goal-oriented abstractions of the task that LLMs are fine-tuned on.
>
> $\color{red}{^4}$ As explained earlier, however, it's entirely possible for a predicate to pertain to the goal-oriented abstractions of Task A yet remain unrelated to Task B (i.e., Task A and B share the same state space but have very different reward functions). Our ablation study confirms that under this scenario, this predicate can be probed from LLMs adapted to Task A but not Task B. Concretely, we have created a new variant of GRIPPER where an originally goal-unrelated predicate (e.g., information of objects' color) pertains to $Q^*$ abstraction. As such, the recovery rate of color information has significantly increased from zero. This further consolidates our conclusion that SFT mainly enhances goal-oriented abstractions. Due to the limited space, we summarize this finding in the main body of the paper (L337-L340) and provide details in Appendix K.

---

> ### Author Response · Authors · 2024-08-07
> **Thank you for your review; we're ready to answer any further questions**
>
> Dear reviewer,
>
> Thank you once again for your review! We really appreciate your recognition of all aspects of our work, including our addressed problem, framework, presentation, and the design of our task and experiments. We believe the major concern might be due to some misunderstandings, and hopefully, our rebuttal has already cleared it up. Meanwhile, we hope that our additional experiments in response to your concerns and suggestions will make you more convinced about the soundness of our findings. Please don't hesitate to let us know if you have any concerns that are unaddressed or if we need to clarify something further. We're always on standby to respond, and we're very looking forward to an engaged and productive discussion!

---

> > ### Comment · Reviewer_4mUW · 2024-08-10
> > **Response to Rebuttal**
> >
> > Thanks for the really well-done rebuttal! Your response clarified a major misunderstanding on my end and thus cleared many of my resulting concerns about your experiments and results. I especially appreciate the added experiment on SFT with sub-optional data following my question. Therefore, I am happy to raise my scores.

---

> > > ### Author Response · Authors · 2024-08-11
> > >
> > > Dear Reviewer,
> > >
> > > We're truly heartened and thrilled to know that our responses are able to address your concerns. This is incredibly important to us! Your interesting question has inspired us to uncover more insights. We will definitely incorporate them in the revised version.

---

### Official Review · Reviewer_ZobB · 2024-07-16

**Soundness:** 3
**Presentation:** 3
**Contribution:** 2
**Rating:** 7
**Confidence:** 4

**Summary:**

In this work, the authors attempt to examine whether large language models (LLMs) possess representations that can work as the world model. To this end, they target different state abstraction levels, world-irrelevant abstraction, $Q^*$-irrelevant abstraction, and $pi^*$-irrelevant abstraction, which are based on state abstraction in reinforcement learning. Their objective is to figure out how much of each state abstraction can be recovered from the last hidden representations of LLMs, and the authors propose a text planning task named *RePlace*. Given the objective, RePlace is designed to have distinct state abstractions for the three abstraction levels as well as the raw state. The authors empirically confirm that both pre-trained LLMs and LLMs fine-tuned on this task fail to recover the world-irrelevant abstraction, whereas the $Q^*$-irrelevant and $pi^*$-irrelevant abstractions exhibit higher recovery rates.

**Post-Rebuttal/Discussion**: I find the author response fair and am raising my score to 7.

**Strengths:**

- Given the extensive use of transformer models these days, establishing sound findings in this regard (whether LLMs can provide representations useful for recovering world dynamics information) can be an important problem.
- The proposed task, RePlace, is carefully designed with different levels of state abstractions in mind. It allows the authors to conduct the empirical analysis that shows the differences in the recovery rates for different state abstractions.

**Weaknesses:**

- The finding that the fine-tuned language models do not preserve the features needed for recovering the world dynamics may not be entirely surprising, as fine-tuning enforces the bias needed for the specific problem to the models and much of irrelevant aspects often end up being ignored.
- In addition to my first point, I believe more interesting findings can be derived with pre-trained LLMs, as there is a possibility that the pre-training provides enough signals to the LLMs to catch the world dynamics information. While the authors do analyze pre-trained LLMs with in-context learning in the proposed benchmark, according to Table 1, their performance is quite low in the first place, which makes it difficult to examine this hypothesis.

**Questions:**

Please take a look at the Weaknesses section above.

**Limitations:**

The authors provide a fair list of limitations in Appendix A.

---

> ### Author Rebuttal · Authors · 2024-08-07
>
> Thank you for your detailed and positive feedback. We address all of your concerns below.
>
> ### **Weakness 1: finding not entirely surprising**
>
> >(**Unsurprising findings**) The finding that the fine-tuned language models do not preserve the features needed for recovering the world dynamics may not be entirely surprising…
>
> * **It's indeed surprising when viewed in the context of existing work**. We provide detailed elaboration in the [Global Rebuttal](https://openreview.net/forum?id=lzfzjYuWgY&noteId=i9JnaUoNpz), with other surprising findings and broader significance of our framework. We sincerely invite you to review it.
>
> > (**It's unsurprising because**) fine-tuning enforces the bias … and much of irrelevant aspects often end up being ignored
>
> * In fact, the **world dynamics is useful for the task**, albeit in a less goal-oriented way. As illustrated in Figure 3, the optimal plan is based on predicting future states. Generally speaking, planning is searching for the shortest action sequences induced by which the predicted future state matches the goal.
> * Nevertheless, **Although $\pi^\*$ abstraction is sufficient for predicting the optimal actions, we still have probed $Q^\*$ abstraction with a high recovery rate** (Figure 5&6, discussed in L324-330). It reveals that LLMs-sft preserves the impact of each possible action, despite being fine-tuned only on next-token prediction. To the best of our knowledge, this interesting finding is novel and can inspire future research to conduct more in-depth analysis.
>
> ### **Weakness 2: Probing with pre-trained LLMs**
>
> > (**Experiments with pre-trained LLMs**) more interesting findings can be derived with pre-trained LLMs … (but) their performance is quite low in the first place, which makes it difficult to examine this hypothesis.
>
> * You are correct that we didn't conduct probing experiments using pre-trained LLMs, given their near-random performance without fine-tuning. Also, it's a common practice [1, 2, 3, 4] that probing from Transformers/LLMs that are trained or fine-tuned on next-token prediction to see if implicit world models emerge, as this is a common interest in the field.
> * To completely address your concern, we conduct new probing experiments on top of Phi3-17b, a brand-new SOTA LLM that achieves 19.94% success rate on GRIPPER without any fine-tuning. From its representations, goal-oriented abstractions can be probed with a significantly higher recovery rate than the world-irrelevant abstraction. We further include two smaller and weaker pre-trained LMs for comparison. The results suggest that more powerful pre-training leads to a higher tendency to prioritize goal-oriented abstractions over a more general one. Please refer to the [Global Rebuttal](https://openreview.net/forum?id=lzfzjYuWgY&noteId=i9JnaUoNpz) for detailed results and in-depth analysis.
>
> [1] Li, Kenneth, et al. "Emergent World Representations: Exploring a Sequence Model Trained on a Synthetic Task." ICLR, 2023.
>
> [2] Li, Belinda Z., Maxwell Nye, and Jacob Andreas. "Implicit Representations of Meaning in Neural Language Models." ACL, 2021.
>
> [3] Hazineh, Dean S., Zechen Zhang, and Jeffery Chiu. "Linear Latent World Models in Simple Transformers: A Case Study on Othello-GPT." arXiv preprint arXiv:2310.07582 (2023).
>
> [4] Kim, Najoung, and Sebastian Schuster. "Entity Tracking in Language Models." ACL, 2023.

---

> ### Author Response · Authors · 2024-08-07
> **Thank you for your review; we're ready to answer any further questions**
>
> Dear reviewer,
>
> Thank you once again for your review! We are truly grateful for your appreciation. We hope our rebuttal clearly explains why our findings are surprising and important and that our additional experiments with pre-trained LLMs yield some other surprising and interesting findings and address all of your concerns. Please don't hesitate to ask any further questions. We are always ready to respond, and we're very looking forward to an engaged and productive discussion!

---

> ### Author Response · Authors · 2024-08-11
>
> Dear Reviewer,
>
> We appreciate the effort you've put into reviewing our paper and offering really helpful feedback. We're wondering if you've had a chance to go through our responses. As the discussion period is nearing its close, we want to ensure that any remaining questions or concerns are fully addressed. If you feel that the main concerns (surprisal of findings and probing with pre-trained LLMs) have been satisfactorily resolved, could you please consider increasing the score to reflect that? We would be truly grateful.

---

### Author Rebuttal · Authors · 2024-08-07

We thank all reviewers for their time and feedback. We are encouraged that all aspects of our work are widely recognized. The reviewers found our addressed problem to be new and important (R1, R2, R3), our framework novel and neat (R2, R4, R5), our task/datasets thoughtfully designed and thus useful for future research (R1, R2, R3, R4, R5), experiments well-designed (R3), findings sound (R1), and our paper well-written (R2, R3, R4, R5). Most of the concerns regard two interrelated aspects, namely the findings' surprisal (R1, R3, R5) and the lack of probing experiments on pre-trained LLMs without ICL/SFT (all reviewers$\color{red}{^1}$). We address both of them below.
### **1. Surprisal of findings**

One primary concern is that the finding, *SFT mainly enhances goal-oriented abstraction*, is not entirely surprising$\color{red}{^2}$. Below, we elaborate on this and other surprising findings, as well as the broader significance of our framework.
* **This finding is surprising when viewed in the context of existing work.** While not entirely unexpected in hindsight, it undermines an increasingly widespread belief that *an implicit world model can emerge from next-token prediction (NTP) training* [1]. This belief is encouraged by recent studies demonstrating that world states can be recovered from Transformer representations [2,3]. When we use a testbed that can differentiate abstraction at various levels, however, we find that LLMs tend to discard the transition dynamics, essential ingredient of world models, when possible. This finding also explains and hence resolves the conflicting conclusions in existing works. For instance, [2,4] successfully probes the Othello board state, which is already the goal-oriented abstraction for predicting legal moves (the task Transformers are trained on), whereas [5] struggles to probe the entity status, which falls outside the goal-oriented abstractions for which LLMs are optimized.
* **It’s more surprising how probing different sets of predicates WITHOUT the notion of world abstractions can lead to directly opposing conclusions**. Imagine two researchers independently assessing an LLM's world representation on RePlace, using current methodologies. As demonstrated in our experiments (Sec 6.2, illustrated in L374-379), Researcher A selecting `agentLoc`, `nearby`, and `held_u`, might infer that SFT enforces LLMs to develop an implicit world model, while Researcher B, focusing on `store` and `boxName`, comes to a totally different conclusion, arguing that LLMs-sft have little awareness of the underlying world. This disparity reflects the current state of the field$\color{red}{^3}$. Thus, our framework provides a more grounded basis for future research, avoiding blind spot conflict and enabling others to dissect and critically evaluate claims in a principled way.

* It's noteworthy that **LLM-sft representations encode the $Q^\*$ abstraction, which could have been abstracted away** as $\pi^\*$ abstraction is sufficient for deriving optimal actions. It reveals that LLMs-sft preserve the long-term impact of actions even when fine-tuned with NTP only. This finding is novel in this area, and it leaves many interesting questions open for future work: How does this abstraction emerge from SFT? Could different training methods enhance or diminish it? Does it affect only the immediate prediction, or do LLMs continuously revisit previous abstractions during decoding?

### **2. Probing experiments on pre-trained LMs**
Similarly, another concern is that *the pre-trained LLMs, especially those that can perform the task well without SFT, may tend to maintain general abstractions, but SFT fundamentally alters this pattern*. To best address it, we employ Phi3-17b [6], a brand-new open LLM that hits a 19.94% success rate on GRIPPER without SFT. To further understand the impact of pre-training on the encoding of world abstractions, we employ two weaker and smaller LLMs, Phi3-3.8b and Pythia-70m, which reach near-random success rates$\color{red}{^4}$, for comparative analysis. We additionally fine-tune another Phi3-17b. Plot A in the **attached pdf** reports the average recovery rate ($\color{red}{\text{RR}}$) of predicates within each type of world abstractions from different LLMs, and Plot B reports the RR of each predicate across all LLMs. We highlight three key findings below.

1) **Pre-trained Phi3-17b prioritizes maintaining goal-oriented abstractions over a more general one**. Plot A clearly shows that. And SFT substantially strengthens this tendency.
2) **As LLMs increase in scale and capability** (Pythia<Phi3-3.8b<Phi3-17b), **they are more likely to maintain goal-oriented abstractions over a more general one**, thereby widening the gap between their RR. This is apparent from Plot B, where the RR variance among different LLMs is mainly found in predicates that pertain to $\pi^*$- and $Q^*$-irrelevant abstractions. In particular, the predicates that uniquely pertain to goal-oriented abstractions, such as `nearby` and `nextObjDirect`, are probed with remarkably higher recovery rates from Phi3-17b than from Pythia. In contrast, the RR of `store` and `held_g`, essential for world-irrelevant abstraction, are almost identical across all LLMs variants. It suggests that **more advanced pre-training doesn't necessarily enhance the encoding of world dynamics**.
3) **Advanced pre-training is NOT sufficient for efficient world modeling**. Interestingly, probing from Phi3-17b has much higher RR of `boxName`, which is neither relevant to task completion nor to the transition dynamics. This verifies our Finding 4 (L366-371) that LLMs are limited in building world representations.

We thank the reviewers for their suggestion of adding this experiment. It turns out to be a good chance to gain new insights about SOTA LLM's world representation which supplement and consolidate our original findings. We will include these results in the revised version, which would be an easy fix.

---

### Author Response · Authors · 2024-08-07
**Footnotes and Reference list for Global Rebuttal**

### **Footnotes**
$\color{red}{^1}$ Although R4 (nz33) didn't straightforwardly suggest conducting experiments with pre-trained LLMs, their recommendation of conducting experiment with another LLM family is also addressed here.

$\color{red}{^2}$ Due to the space limits, we didn't thoroughly discuss this aspect in the paper. We'll add this discussion in the revised version.

$\color{red}{^3}$ **While some studies continue to produce "surprising" findings suggesting that Transformers/LLMs trained or fine-tuned on next-token prediction develop internal world models** [2,3,4,7], **other research provides counter-evidence, cautioning that the capabilities of world modeling are overestimated** [5,8,9].

$\color{red}{^4}$ Phi3-3.8b and Pythia-70 achieve success rates of approximately 4% and 1%, respectively, on GRIPPER .

### **Reference**

[1] Kenneth Li, "Do Large Language Models learn world models or just surface statistics?", The Gradient, 2023.

[2] Li, Kenneth, et al. "Emergent World Representations: Exploring a Sequence Model Trained on a Synthetic Task." ICLR, 2023.

[3] Li, Belinda Z., Maxwell Nye, and Jacob Andreas. "Implicit Representations of Meaning in Neural Language Models." ACL, 2021.

[4] Hazineh, Dean S., Zechen Zhang, and Jeffery Chiu. "Linear Latent World Models in Simple Transformers: A Case Study on Othello-GPT." arXiv preprint arXiv:2310.07582 (2023).

[5] Kim, Najoung, and Sebastian Schuster. "Entity Tracking in Language Models." ACL, 2023.

[6] Abdin, Marah, et al. "Phi-3 technical report: A highly capable language model locally on your phone." arXiv preprint arXiv:2404.14219 (2024).

[7] Gurnee, Wes, and Max Tegmark. "Language Models Represent Space and Time." ICLR, 2024.

[8] Altmeyer, Patrick, et al. "Position Paper: Against Spurious Sparks-Dovelating Inflated AI Claims." ICML, 2024.

[9] Wang, Ruoyao, et al. "Can Language Models Serve as Text-Based World Simulators?." ACL, 2024.

---

### Decision · Program_Chairs · 2024-09-25

**Decision:**

Accept (poster)

**Comment:**

This paper emphasizes the importance of considering different scopes of state abstraction when analyzing internal representations of language models to determine whether they contain world models. The paper demonstrates these results by analyzing representations of several fine-tuned LMs on two synthetic planning domains, and finds that their representations are goal-oriented. After discussion the reviewers generally agree that the revised version of the paper will merit acceptance. There were some remaining concerns about the generality of the result, including the focus on synthetic tasks, the range of models considered, and most centrally the need to fine-tune which will alter the representation content. These concerns limit the paper from addressing the broadest form of the questions posed, but the reviewers do agree that the revised version will addresses some aspects of the questions and makes a useful contribution in doing so.